# ♥️FREPHYS: FREQUENCY-AWARE DIFFUSION MODEL FOR REMOTE PHYSIOLOGICAL MEASUREMENT

## ABSTRACT

Remote photoplethysmography (rPPG) enables non-contact physiological monitoring by capturing subtle skin color variations in facial videos. Existing approaches predominantly rely on time-domain modeling to extract cardiac-related periodic signals, but they are highly vulnerable to motion artifacts and illumination changes, where physiological clues are easily obscured by noise. To address these challenges, we propose a **Fre**quency-aware **Phys**iological diffusion model, dubbed **FrePhys**, that integrates physiological frequency priors into rPPG estimation. Specifically, it first employs a *physiological bandpass filter* to suppress out-of-band noise, followed by *physiological spectrum modulation* and *adaptive spectrum selection* for in-band noise suppression and pulse-related clues enhancement. A *cross-domain representation learning* module then fuses frequency-domain insights with the deep time-domain features to capture spatial–temporal dependencies. Finally, a frequency-aware conditional diffusion process iteratively reconstructs high-fidelity rPPG signals. Extensive experiments on multiple datasets demonstrate that our method significantly outperforms existing state-of-the-art methods, particularly under challenging motion conditions, highlighting the effectiveness of incorporating frequency priors. The source code is available at https://anonymous.4open.science/r/FrePhys.

## 1 INTRODUCTION

Physiological signals, such as heart rate (HR), heart rate variability (HRV), and respiration frequency (RF), are essential indicators of physical and mental health. Traditional electrocardiogram (ECG) and photoplethysmography (PPG) methods require the use of skin-contact devices, which can cause discomfort and inconvenience to subjects. Recently, remote photoplethysmography (rPPG) (Verkruysse, 2008) has emerged as a promising non-invasive optical technique, enabling applications in health monitoring (Huang et al., 2023), face anti-spoofing (Yu et al., 2021), and psychological stress assessment (Gedam & Paul, 2020), among others. However, a key challenge for rPPG remains how to accurately capture subtle skin color changes caused by blood volume fluctuations in facial videos recorded by ordinary cameras.

Early rPPG research (Verkruysse, 2008; Poh et al., 2010; De Haan & Jeanne, 2013; Wang et al., 2016) mainly relied on traditional signal processing methods to recover subtle rPPG signals, which are often limited to specific signal assumptions. Recently, the emerging development of deep learning has fostered numerous sophisticated deep rPPG models (Yu et al., 2019; Niu et al., 2020; Lu et al., 2021; Liu et al., 2023; Qian et al., 2024a; Zou et al., 2025b). While these models perform well in controlled environments, they are often limited in robustness to noisy conditions such as motion and illumination (Qian et al., 2025; Shao et al., 2025). To alleviate this problem, the denoising diffusion probabilistic models (DDPMs) have been introduced for rPPG estimation (Chen et al., 2024; Qian et al., 2025), due to their remarkable capacity for modeling noise distributions and recovering clean signals from heavily corrupted observations. However, these pioneering efforts were primarily conducted in the time domain, where noise often exhibits irregular and chaotic patterns, as shown in the upper part of Fig. 1(b) and (c), making it challenging to separate physiological components from noise. Several recent studies have begun to exploit frequency information in rPPG. The most common usage is to impose auxiliary frequency-based losses during training (Yu et al., 2023; Sun & Li, 2024; Zou et al., 2025b). Other approaches employ Fourier transform blocks to enhance feature

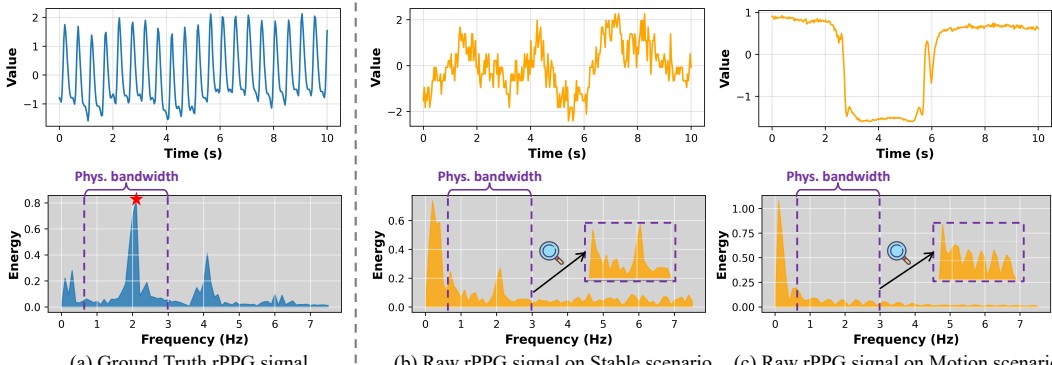

Figure 1: Visualization of the differences between Ground-truth and Raw rPPG signals in time and frequency domains. (a) Ground-truth rPPG signal, where the spectrum exhibits clear physiological priors with a dominant peak (marked by the ★) corresponding to HR, obtained by multiplying the frequency by 60. (b-c) Raw rPPG signals extracted from facial videos under stable and motion conditions by averaging green-channel pixel intensities over time (Wang et al., 2016).

representation (Zou et al., 2025b) or synthesize negative samples (Yue et al., 2023). While these methods demonstrate the utility of frequency information, they fall short of fully leveraging the inherent physiological priors in the frequency domain and largely overlook the distinct challenges introduced by motion-induced noise.

In this work, we take a closer look at the frequency domain and show that rPPG signals, driven by cardiac rhythms, are quasi-periodic and exhibit clear frequency-domain priors (Gideon & Stent, 2021; Speth et al., 2023), as illustrated in Fig. 1(a): (*i*) **Physiological Band Constraint:** spectral energy is mainly concentrated within a fixed physiological band, typically [0.66, 3.0] Hz, corresponding to the normal HR range; (*ii*) **Dominant Peak Property:** a strong spectral peak emerges within this band, reflecting the periodic cardiac rhythm, while other in-band noise frequencies carry relatively low energy. For an intuitive illustration, we further visualize raw rPPG signals from facial videos under both stable and motion conditions by computing the mean green-channel intensity over time (Wang et al., 2016), as shown in Fig. 1(b)(c). In the time domain, noise is heavily entangled with the signal, making separation very difficult. By contrast, the frequency domain reveals two distinct categories of noise: out-of-band components and residual in-band noise, where most noise energy is concentrated on low-frequency components outside the physiological band. The stable conditions exhibit a clear spectral peak within the physiological band, while motion disperses the in-band energy, thereby complicating the denoising process. These observations naturally motivate our central question: ***How to suppress both out-of-band and in-band noise while effectively preserving physiologically meaningful spectral information?***

To address these challenges, we propose a novel frequency-aware physiological diffusion model, FrePhys, whose key idea is to incorporate physiological frequency priors into the denoising process, thereby combining spectral priors with temporal dynamics for more effective rPPG signal recovery. To suppress out-of-band noise, we design a *physiological bandpass filter* that preserves only the physiological frequency range. To further handle in-band noise while emphasizing meaningful spectral information, we introduce a *physiological spectrum modulation* to enhance true cardiac harmonics and an *adaptive spectrum selection* to dynamically suppress residual components. Furthermore, we introduce a *cross-domain representation learning* module, which leverages cross-attention to fuse spectral priors with temporal representations, thereby guiding the denoising process with both frequency-domain regularities and temporal dynamics.

**Contribution Summary**: (*i*) We highlight the importance of explicitly leveraging physiological frequency priors for robust rPPG estimation. (*ii*) We propose FrePhys, a frequency-aware diffusion framework that integrates physiological frequency denoise with cross-domain representation learning. (*iii*) Unlike previous diffusion-based methods that operate purely in the time domain, our model incorporates frequency-domain conditioning to better capture the quasi-periodic nature of rPPG. (*iv*) Extensive experiments on four public benchmarks show that our method achieves state-of-the-art performance in both accuracy and robustness.

## 2 RELATED WORK

Early rPPG estimation methods primarily relied on signal processing techniques, such as GREEN (Verkruysse, 2008), ICA (Poh et al., 2010), CHROM (De Haan & Jeanne, 2013), and POS (Wang et al., 2016). With the rise of deep learning, a wide range of models have been introduced, including CNN-based methods (e.g., DeepPhys (Chen & McDuff, 2018), PhysNet (Yu et al., 2019), CVD (Niu et al., 2020), TS-CAN (Liu et al., 2020)), Transformer-based methods (e.g., PhysFormer (Yu et al., 2022), EfficientPhys (Liu et al., 2023), Dual-TL (Qian et al., 2024a)), and Mamba-based methods (e.g., PhysMamba (Luo et al., 2024), RhythmMamba (Zou et al., 2025b)), Diffusion-based methods (e.g., DiffPhys (Chen et al., 2024), PhysDiff (Qian et al., 2025)) etc. Beyond purely temporal modeling, several works have attempted to incorporate frequency-domain information, including frequency-aware loss functions (e.g., PhysFormer (Yu et al., 2022), Contrast-Phys (Sun & Li, 2024)) or frequency representation learning (e.g., Yue et al. (Yue et al., 2023), RhythmMamba (Zou et al., 2025b)). A more detailed discussion is provided in Appendix A.

**Remark**. Our approach fundamentally rethinks how frequency information is used in rPPG. Most previous works leverage frequency only as an offline post-processing tool for HR computation (Niu et al., 2020; Lu et al., 2021; Yu et al., 2022; Liu et al., 2023; Qian et al., 2024a), or as auxiliary modules such as Fourier transform blocks for representation enhancement (Zou et al., 2025b), negative sample synthesis (Yue et al., 2023), or auxiliary frequency losses during training (Yu et al., 2023; Sun & Li, 2024; Zou et al., 2025b). In contrast, we directly integrate physiological frequency priors into the diffusion model through a three-stage filtering mechanism: suppressing out-of-band noise, enhancing true cardiac harmonics, and adaptively removing in-band residual noise. Moreover, instead of limiting frequency regulation to training, we enforce frequency-aware denoising at every diffusion step, during both training and inference. To the best of our knowledge, this is the first work to seamlessly embed physiological frequency priors into diffusion modeling for rPPG, leading to robust and high-fidelity signal reconstruction under challenging noise conditions.

## 3 METHODOLOGY

Remote physiological measurement from facial videos can be regarded as a video sequence to signal sequence problem. Let $\mathbf{V} \in \mathbb{R}^{T \times 3 \times H \times W}$ denote a raw facial video clip containing $T$ frames with 3 color channels and spatial resolution $H \times W$. Following established rPPG preprocessing protocols (Niu et al., 2020; Qian et al., 2025), we extract $N$ facial regions of interest (ROIs) through landmark alignment and pixel-level average pooling, constructing a multi-scale temporal map (MSTmap) $\mathbf{X} \in \mathbb{R}^{T \times N \times C}$ as the model input. The objective is to recover the clean periodic rPPG signal $\mathbf{Y} \in \mathbb{R}^{T}$ from $\mathbf{X}$, formulated as learning a denoising function $f_\theta : \mathbf{X} \mapsto \mathbf{Y}$, where $\theta$ denotes trainable parameters. In this work, we propose a novel physiological frequency-aware diffusion model to consider the important clues from the physiological frequency domain. The overview of our method is illustrated in Fig. 2, where the details are described as follows.

### 3.1 PHYSIOLOGICAL FREQUENCY DENOISER

**Physiological Bandpass Filter (PBF).** Inspired by the fact that true cardiac activities mainly fall within a fixed frequency bandwidth, typically [0.66,3.0] Hz (Wang et al., 2016), we devise a *Physiological Bandpass Filter* that directly isolates cardiac frequency components in the spectral space. Specifically, we first project the frequency condition $\mathbf{C}^\mathbf{P} \in \mathbb{R}^{T \times N \times C}$ into a $D$-dimensional latent space $\mathbf{Z} \in \mathbb{R}^{T \times N \times D}$, then transform it to frequency domain via the Discrete Fourier Transform $\mathcal{F}$,

$$\mathbf{Z}^{F'} = \mathcal{F}(\mathbf{Z}) \in \mathbb{C}^{(\lfloor T/2 \rfloor + 1) \times N \times D}. \tag{1}$$

Then, the noisy frequency components outside the physiological bandwidth range can be discarded using an ideal band-pass filter:

$$\mathbf{Z}^F = \mathbf{Z}^{F'} \odot f(\lambda_i), \quad \text{for } i = 0, \dots, \lfloor T/2 \rfloor \tag{2}$$

where $\odot$ denotes the Hadamard product. $\lambda_i$ denotes the physical frequency corresponding to the $i$-th frequency bins and $\lambda_i = i f_s / T$ Hz, with $f_s$ denoting the sampling rate. $f(\lambda_i)$ is an indicator function, which outputs 1 when $(0.66 \leq \lambda_i \leq 3.0)$ and 0 otherwise.

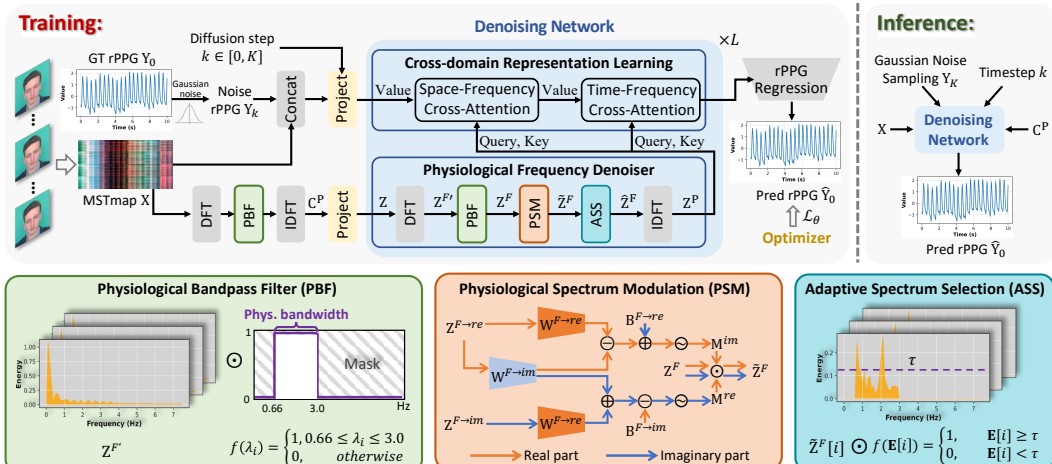

Figure 2: The pipeline of **FrePhys** is implemented by the frequency-aware diffusion model. Given a facial video, we first construct MSTmap $\mathbf{X}$ as the temporal condition and generate the frequency condition $\mathbf{C}^{\mathbf{P}}$ by applying the PBF. During training, we initially generate noise rPPG $\mathbf{Y}_k$ by adding Gaussian noise to Ground Truth rPPG $\mathbf{Y}_0$ for the $k$-th step. Then, we input $\mathbf{Y}_k$, $\mathbf{X}$, $k$, and $\mathbf{C}^{\mathbf{P}}$ into the *Denoising Network*. Specifically, the frequency condition $\mathbf{C}^{\mathbf{P}}$ is fed into the *Physiological Frequency Denoiser* module to enhance physiological spectral clues through three key steps: (*i*) PBF removes out-of-band noise based on the physiological frequency bandwidth [0.66,3.0] Hz; (*ii*) PSM emphasizes valid physiological harmonics by modeling interactions between real and imaginary components; (*iii*) ASS dynamically suppresses in-band noise using data-driven energy thresholds. Next, with *Cross-domain Representation Learning*, our **FrePhys** includes frequency-domain denoised information into space and time dependencies modeling to estimate the high-fidelity rPPG signal. During inference, the initial rPPG $\mathbf{Y}_K$ is randomly sampled from Gaussian noise, with frequency condition and denoising network processes mirroring those used in training.

**Physiological Spectrum Modulation (PSM).** While the *Physiological Bandpass Filter* is effective in removing noise outside the physiological frequency band, another challenge still exists where noise components may overlap or closely resemble physiological signals within this band identified by PBF. Such overlapping frequencies may severely distort the signal, making it difficult to accurately extract physiological features. We apply a learnable *Physiological Spectrum Modulation* in the frequency domain to emphasize true physiological harmonics while suppressing non-physiological components. Specifically, given the physiological frequency representation $\mathbf{Z}^F \in \mathbb{C}^{(\lfloor T/2 \rfloor+1) \times N \times D}$, we denote its real and imaginary parts as $\mathbf{Z}^{F \to re}$ and $\mathbf{Z}^{F \to im}$, separately. To achieve more exhaustive spectrum modulation, we encode the real and imaginary parts separately to generate the modulation signals, which are formulated as:

$$\mathbf{M} = \sigma(\mathbf{Z}^F \mathbf{W}^F + \mathbf{b}^F), \tag{3}$$

where $\sigma$ is the ReLU activation function, $\mathbf{W}^F = (\mathbf{W}^{F \to re} + j \cdot \mathbf{W}^{F \to im}) \in \mathbb{R}^{D \times D}$ is the trainable complex number weight matrix with $\{\mathbf{W}^{F \to re}, \mathbf{W}^{F \to im}\} \in \mathbb{C}^{D \times D}$, and $\mathbf{b}^F = (\mathbf{b}^{F \to re} + j \cdot \mathbf{b}^{F \to im}) \in \mathbb{C}^D$ is the trainable complex number biases with $\{\mathbf{b}^{F \to re}, \mathbf{b}^{F \to im}\} \in \mathbb{C}^D$. According to the rule of multiplication of complex numbers (details can be seen in Appendix B.1), we further unfold into real and imaginary parts as follows:

$$\begin{aligned} \mathbf{M}^{re} &= \sigma(\mathbf{Z}^{F \to re} \mathbf{W}^{F \to re} - \mathbf{Z}^{F \to im} \mathbf{W}^{F \to im} + \mathbf{b}^{F \to re}), \\ \mathbf{M}^{im} &= \sigma(\mathbf{Z}^{F \to im} \mathbf{W}^{F \to im} + \mathbf{Z}^{F \to re} \mathbf{W}^{F \to re} + \mathbf{b}^{F \to im}). \end{aligned} \tag{4}$$

Afterwards, the generated complex signal is used to modulate counterparts of the original frequency-domain feature, which can be written as,

$$\tilde{\mathbf{Z}}^F = \mathbf{M} \odot \mathbf{Z}^F \in \mathbb{C}^{(\lfloor T/2 \rfloor+1) \times D}. \tag{5}$$

By Theorem 1, this spectral multiplication operation in Eq. 5 is mathematically equivalent to a global circular convolution in time, endowing each sequence with a content-adaptive receptive field that is

ideal for capturing periodic cardiac rhythms. By means of Eq. 5, the physiological frequency components can be effectively enhanced via direct spectral modulation. The detailed proof of *theorem* 1 is illustrated in Appendix C.1.

**Theorem 1** *(Frequency-domain Convolution Theorem) The multiplication of two signals in the frequency domain is equivalent to the frequency transformation of a circular convolution of these two signals in the temporal domain, which can be summarized as:*

$$\mathcal{F}[\mathbf{M}(v) \otimes \mathbf{Z}(v)] = \mathcal{F}(\mathbf{M}(v)) \odot \mathcal{F}(\mathbf{Z}(v)), \tag{6}$$

*where $\otimes$ and $\odot$ represent the circular convolutional operation and element-multiplication operation, respectively, $\mathbf{M}(v)$ and $\mathbf{Z}(v)$ represent two signals for the time variable $v$, and $\mathcal{F}(\cdot)$ denotes the Discrete Fourier Transform.*

**Adaptive Spectrum Selection (ASS).** While the physiological spectrum modulation block amplifies the cardiac band, some noise components whose frequencies lie close to the physiological range may remain. To robustly isolate the true pulse periodicity, we further introduce an *Adaptive Spectrum Selection* block that learns a data-driven threshold $\tau$ in the frequency domain, retaining dominant spectral components and discarding residual noise. Concretely, for modulated physiological spectrum $\widetilde{\mathbf{Z}}^F$, we first calculate its per-frequency energy:

$$\mathbf{E}[i] = \sqrt{(\widetilde{\mathbf{Z}}^{F \to re}[i])^2 + (\widetilde{\mathbf{Z}}^{F \to im}[i])^2} = \left\| \widetilde{\mathbf{Z}}^F[i] \right\|_2, \quad i = 0, 1, \dots, \lfloor T/2 \rfloor. \tag{7}$$

Then we employ a learnable threshold $\tau$ to discern between cardiac pulse and potential noise. We formulate this adaptive thresholding as follows:

$$\hat{\mathbf{Z}}^F[i] = \widetilde{\mathbf{Z}}^F[i] \odot f(\mathbf{E}[i]), \tag{8}$$

here, $f(\mathbf{E}[i])$ is a binary mask where frequencies with energy above the threshold ($\mathbf{E}[i] \geq \tau$) are retained, and others are filtered out. Finally, an inverse DFT restores the time-domain signal containing only the dominant cardiac activity:

$$\mathbf{Z}^{\mathbf{P}} = \mathcal{F}^{-1}(\hat{\mathbf{Z}}^F). \tag{9}$$

### 3.2 Cross-domain Representation Learning

Time domain modeling focuses on local dependencies and transient behaviors, while frequency domain analysis provides insights into the global correlations and periodicity of the data. Therefore, combining these two domains is a promising approach to recover high-fidelity rPPG signals. To effectively integrate intermediate representations $\mathbf{Z}$ in the time domain with frequency-domain priors $\mathbf{Z}^{\mathbf{P}}$, we propose a cross-domain representation learning module. Specifically, we perform $L$ alternating cross-attention layers that can progressively learn the various input domains for representation learning. In each layer $l$, the initial physiological frequency representation is first obtained by applying PBF, followed by PSM and ASS modules. Next, the physiological frequency representation $\mathbf{Z}^{\mathbf{P}}$ and intermediate representations $\mathbf{Z}$ are integrated by cross-attention across the spatial and temporal axis, respectively. Formally, this process can be formulated as follows:

$$\begin{aligned}
\mathbf{Z}^{\mathbf{P},(l)} &= \text{ASS}(\text{PSM}(\text{PBF}(\mathbf{Z}^{(l)}))), \\
\mathbf{Z}^{(l)'} &= \text{CA}(\mathbf{Z}^{\mathbf{P},(l)}, \mathbf{Z}^{(l)}) + \mathbf{Z}^{(l)}, \\
\mathbf{Z}^{(l+1)} &= \text{CA}(\mathbf{Z}^{\mathbf{P},(l)}, \mathbf{Z}^{(l)'}) + \mathbf{Z}^{(l)'},
\end{aligned} \tag{10}$$

where $\text{CA}(a, b)$ refers to Cross-Attention, with $a$ denotes query and key, and $b$ denotes value.

### 3.3 Frequency-aware Diffusion Model

Recently, denoising diffusion probabilistic models (DDPMs) (Ho et al., 2020) have emerged as powerful generative frameworks that progressively refine noisy inputs through learned reverse Markov chains, capturing complex data distributions. Inspired by this, some diffusion-based methods (Qian et al., 2025; Chen et al., 2024) for rPPG estimation have been proposed and achieved SOTA performance. They treat the rPPG estimation task as calculating the conditional rPPG signal probability

distribution $q(\mathbf{Y}_0|\mathbf{C})$, where $q(\mathbf{Y}_0)$ is the clean rPPG distribution, and the condition $\mathbf{C}$ for probability distribution calculation is generally the input $\mathbf{X}$ in the time domain. However, these diffusion models mainly focus on time-domain conditioning and overlook the unique spectral prior.

To alleviate this limitation, we introduce a novel frequency-aware diffusion model that explicitly incorporates physiological frequency priors to guide the generation of high-fidelity rPPG signals. Specifically, our frequency-aware diffusion model fuses with physiological frequency condition $\mathbf{C^P}$ to learn the conditional rPPG distribution $q(\mathbf{Y}_0|\mathbf{Y}, \mathbf{X}, \mathbf{C^P})$, through two Markov chain processes of diffusion step $K$, i.e., the forward process and the reverse process.

**Forward Process.** The forward process $q$ incrementally adds Gaussian noise to the ground truth rPPG signal $\mathbf{Y}_0 \in \mathbb{R}^T$ via a fixed Markov chain $\mathbf{Y}_0, \ldots \mathbf{Y}_K$ as follows:

$$q(\mathbf{Y}_{1:K}|\mathbf{Y}_0) = \prod_{k=1}^{K} q(\mathbf{Y}_k|\mathbf{Y}_{k-1}), \quad q(\mathbf{Y}_k|\mathbf{Y}_{k-1}) = \mathcal{N}(\mathbf{Y}_k; \sqrt{1-\beta_k}\mathbf{Y}_{k-1}, \beta_k\mathbf{I}), \qquad (11)$$

where $\beta_k$ is a noise schedule, satisfying $\beta_k < \beta_{k-1}$. As $K$ becomes large, $\mathbf{Y}_K \approx \mathcal{N}(0, \mathbf{I})$.

Following DDPM (Ho et al., 2020), we sample $\mathbf{Y}_k$ from $\mathbf{Y}_0$ at any time step $k$ in a closed form:

$$q(\mathbf{Y}_k|\mathbf{Y}_0) = \mathcal{N}(\mathbf{Y}_k; \sqrt{\bar{\alpha}_k}\mathbf{Y}_0, (1-\bar{\alpha}_k)\mathbf{I}), \qquad (12)$$

where $\alpha_k = 1 - \beta_k$ and $\bar{\alpha}_k = \prod_{s=0}^{k} \alpha_s$. Utilizing the parameterization trick (Kingma & Welling, 2013), we express $\mathbf{Y}_k$ as:

$$\mathbf{Y}_k = \sqrt{\bar{\alpha}_k}\mathbf{Y}_0 + \sqrt{1-\bar{\alpha}_k}\epsilon, \qquad (13)$$

where $\epsilon \sim \mathcal{N}(0, \mathbf{I})$. The detailed derivations are provided in Appendix D.1.

**Reverse Process.** The reverse process aims to estimate the posterior $q(\mathbf{Y}_{k-1}|\mathbf{Y}_k)$. Different from PhysDiff (Qian et al., 2025), in our frequency-aware diffusion model, this distribution is approximated by a neural network $f_\theta$ conditioned on both the time-domain condition $\mathbf{X}$ and the physiological frequency condition $\mathbf{C^P}$:

$$p_\theta(\mathbf{Y}_{k-1}|\mathbf{Y}_k, \mathbf{X}, \mathbf{C^P}) = \mathcal{N}(\mathbf{Y}_{k-1}; \mu_\theta(\mathbf{Y}_k, \mathbf{X}, \mathbf{C^P}, k), \Sigma_\theta). \qquad (14)$$

Next, we show that incorporating the physiological frequency prior $\mathbf{C^P}$ can effectively reduce the uncertainty in the reverse diffusion process, leading to more accurate rPPG signal reconstruction. It can be formalized in Proposition 1.

**Proposition 1** *The conditional entropy is satisfied:*

$$\mathbf{H}(\mathbf{Y}_{k-1}|\mathbf{Y}_k, \mathbf{X}, \mathbf{C^P}) < \mathbf{H}(\mathbf{Y}_{k-1}|\mathbf{Y}_k, \mathbf{X}), \qquad (15)$$

*indicating that the inclusion of additional physiological frequency condition $\mathbf{C^P}$ in the reverse process reduces uncertainty. The detailed proof is provided in Appendix C.2.*

**Accelerated Training.** To enhance the efficiency of our model, we accelerate both the training and sampling processes. Traditional DDPM-based training involves learning to predict the added Gaussian noise at each diffusion step, which can be inefficient (Ho et al., 2020). Instead, our denoising network $f_\theta$ is designed to directly reconstruct the clean rPPG signal $\mathbf{Y}_0$ from the noisy input $\mathbf{Y}_0$, conditioned on $\mathbf{X}, \mathbf{C^P}$, and the timestep $k$:

$$\hat{\mathbf{Y}}_0 = f_\theta(\mathbf{Y}_k, \mathbf{X}, \mathbf{C^P}, k). \qquad (16)$$

In practice, for Equation 14, the mean $\mu_\theta$ and covariance $\sigma_k^2$ in reverse process are parameterized as $\mu_\theta(\mathbf{Y}_k, \mathbf{X}, \mathbf{C^P}, k) = \frac{\sqrt{\bar{\alpha}_k}(1-\bar{\alpha}_{k-1})}{1-\bar{\alpha}_k}\mathbf{Y}_k + \frac{\sqrt{\bar{\alpha}_{k-1}}\beta_k}{1-\bar{\alpha}_k}f_\theta(\mathbf{Y}_k, \mathbf{X}, \mathbf{C^P}, k)$, and $\Sigma_\theta = \sigma_k^2\mathbf{I}$, where $\sigma_k^2 = \frac{1-\bar{\alpha}_{k-1}}{1-\bar{\alpha}_k}\beta_k$. The mathematical details are presented in Appendix D.2. Furthermore, inspired by the Fourier-based loss term, which is beneficial for the accurate reconstruction of the signals(Fons et al., 2022), we propose to guide the diffusion training by applying it to the frequency domain with the Fourier transform. Formally, our training objective integrates both time and frequency-domain constraints:

$$\mathcal{L}_\theta(\hat{\mathbf{Y}}_0, \mathbf{Y}_0) = \underbrace{1 - Pearson(\hat{\mathbf{Y}}_0, \mathbf{Y}_0)}_{\text{time-domain loss}} + \underbrace{MSE(\mathcal{F}(\hat{\mathbf{Y}}_0), \mathcal{F}(\mathbf{Y}_0))}_{\text{frequency-domain loss}}, \qquad (17)$$

where $Pearson$ represents Pearson correlation coefficient, $MSE$ denotes Mean Square Error, and $\mathcal{F}$ denotes the Discrete Fourier Transform.

For inference, we start from $\mathbf{Y}_K \sim \mathcal{N}(0, \mathbf{I})$, $K$, $\mathbf{X}$, and $\mathbf{C^P}$. Then, we follow DDIM (Song et al., 2021a; Qian et al., 2025) and perform the reverse process to obtain the final rPPG signal.

Table 1: Intra-dataset HR estimation results of models on the UBFC-rPPG, PURE, VIPL-HR, and MMPD datasets. **bold**: best results.

| Method | Venue | UBFC-rPPG | | | PURE | | | MMPD | | | VIPL-HR | | |
|---|---|---|---|---|---|---|---|---|---|---|---|---|---|
| | | MAE↓ | RMSE↓ | ρ↑ | MAE↓ | RMSE↓ | ρ↑ | MAE↓ | RMSE↓ | ρ↑ | MAE↓ | RMSE↓ | ρ↑ |
| DeepPhys (Chen & McDuff, 2018) | ECCV'18 | 2.90 | 3.63 | - | 0.83 | 1.54 | 0.99 | 22.27 | 28.92 | -0.03 | 11.0 | 13.8 | 0.72 |
| PhysNet (Yu et al., 2019) | BMVC'19 | 2.95 | 3.67 | 0.97 | 2.10 | 2.60 | 0.99 | 4.80 | 11.80 | 0.60 | 10.80 | 14.80 | 0.20 |
| CVD (Niu et al., 2020) | ECCV'20 | 2.19 | 3.12 | 0.99 | 1.29 | 2.01 | 0.98 | - | - | - | 5.02 | 7.97 | 0.79 |
| TS-CAN (Liu et al., 2020) | NeurIPS'20 | 1.70 | 2.72 | 0.99 | 2.48 | 9.01 | 0.92 | 9.71 | 17.22 | 0.44 | - | - | - |
| Gideon et al.(Gideon & Stent, 2021) | ICCV'21 | 1.85 | 4.28 | 0.93 | 2.30 | 2.90 | 0.99 | - | - | - | 9.01 | 14.02 | 0.58 |
| Dual-GAN (Lu et al., 2021) | CVPR'21 | 0.44 | 0.67 | 0.99 | 0.82 | 1.31 | 0.99 | - | - | - | 4.93 | 7.68 | 0.81 |
| PhysFormer (Yu et al., 2022) | CVPR'22 | 0.50 | 0.71 | 0.99 | 1.10 | 1.75 | 0.99 | 11.99 | 18.41 | 0.18 | 4.97 | 7.79 | 0.78 |
| EfficientPhys (Liu et al., 2023) | WACV'23 | 1.14 | 1.81 | 0.99 | - | - | - | 13.47 | 21.32 | 0.21 | - | - | - |
| Li et al.(Li & Yin, 2023) | ICCV'23 | 0.48 | 0.64 | **1.00** | 0.64 | 1.16 | 0.99 | - | - | - | 4.97 | 7.79 | 0.78 |
| Yue et al.(Yue et al., 2023) | TPAMI'23 | 0.58 | 0.94 | 0.99 | 1.23 | 2.01 | 0.99 | - | - | - | - | - | - |
| Contrast-Phys+(Sun & Li, 2024) | TPAMI'24 | **0.21** | 0.80 | 0.99 | 0.48 | 0.98 | 0.99 | - | - | - | - | - | - |
| DiffPhys (Chen et al., 2024) | Bioeng.'24 | 1.05 | 1.63 | 0.99 | 1.46 | 5.88 | 0.90 | - | - | - | - | - | - |
| CodePhys (Chu et al., 2025) | JBHI'25 | 0.21 | 0.26 | 0.99 | 0.39 | 0.83 | 0.99 | - | - | - | 4.27 | 7.11 | 0.81 |
| RhythmMamba (Zou et al., 2025b) | AAAI'25 | 0.50 | 0.75 | 0.99 | 0.23 | 0.34 | 0.99 | **3.16** | 7.27 | 0.84 | 4.30 | 7.49 | 0.81 |
| PhysDiff (Qian et al., 2025) | AAAI'25 | 0.33 | 0.57 | **1.00** | 0.29 | 0.54 | **1.00** | 7.17 | 9.63 | 0.71 | 3.92 | 6.65 | 0.85 |
| **FrePhys (Ours)** | - | 0.24 | **0.53** | **1.00** | **0.17** | **0.25** | **1.00** | 4.20 | **6.78** | **0.86** | **3.79** | **6.34** | **0.86** |

Table 2: HRV and RF estimation results of models on the UBFC-rPPG dataset. LF, HF, and RF represent low frequency, high frequency, and respiration frequency, respectively. "n.u." denotes normalized units.

| Method | Venue | LF (n.u.) | | | HF (n.u) | | | LF/HF | | | RF (Hz) | | |
|---|---|---|---|---|---|---|---|---|---|---|---|---|---|
| | | SD↓ | RMSE↓ | ρ↑ | SD↓ | RMSE↓ | ρ↑ | SD↓ | RMSE↓ | ρ↑ | SD↓ | RMSE↓ | ρ↑ |
| CVD (Niu et al., 2020) | ECCV'20 | 0.053 | 0.056 | 0.740 | 0.053 | 0.065 | 0.740 | 0.169 | 0.168 | 0.812 | 0.017 | 0.018 | 0.252 |
| Dual-GAN (Lu et al., 2021) | CVPR'21 | 0.034 | 0.035 | 0.891 | 0.034 | 0.034 | 0.891 | 0.131 | 0.136 | 0.881 | 0.010 | 0.010 | 0.395 |
| Gideon et al. (Gideon & Stent, 2021) | ICCV'21 | 0.091 | 0.139 | 0.694 | 0.091 | 0.139 | 0.694 | 0.525 | 0.691 | 0.684 | 0.061 | 0.098 | 0.103 |
| Contras-Phys (Sun & Li, 2022) | ECCV'22 | 0.050 | 0.098 | 0.798 | 0.050 | 0.098 | 0.798 | 0.205 | 0.395 | 0.782 | 0.055 | 0.083 | 0.347 |
| Contrast-Phys+ (Sun & Li, 2024) | TPAMI'24 | 0.025 | 0.025 | 0.947 | 0.025 | 0.025 | 0.947 | **0.064** | **0.066** | 0.963 | 0.029 | 0.029 | 0.803 |
| PhysDiff (Qian et al., 2025) | AAAI'25 | 0.029 | 0.022 | 0.978 | 0.016 | 0.022 | 0.978 | 0.079 | **0.066** | 0.979 | **0.006** | 0.007 | 0.811 |
| **FrePhys (Ours)** | - | **0.016** | **0.014** | **0.988** | **0.013** | **0.014** | **0.988** | 0.079 | **0.066** | **0.989** | **0.006** | **0.005** | 0.845 |

## 4 EXPERIMENTS

### 4.1 EXPERIMENTAL SETUP

**Datasets.** Following (Zou et al., 2025b; Qian et al., 2025), we evaluate our method on four benchmark datasets: UBFC-rPPG (Bobbia et al., 2019) and PURE (Stricker et al., 2014) are two small-scale datasets that contain 59 and 42 videos, respectively, in relatively constrained conditions. MMPD (Tang et al., 2023) dataset is a medium-scale dataset containing 660 videos under 4 distinct lighting configurations. VIPL-HR (Niu et al., 2019) dataset is a large-scale dataset containing 2,378 videos captured across 9 scenarios and 4 recording devices. The detailed description of datasets is provided in Appendix F.1.

**Evaluation Metrics.** We employ three commonly used metrics, Mean Absolute Error (MAE), Root Mean Square Error (RMSE), and Pearson's correlation coefficient $\rho$, to evaluate the performance of HR estimation. For the evaluation of HRV features, we follow the approach outlined in (Yu et al., 2023; Sun & Li, 2024) and employ the Standard Deviation (SD), RMSE, and $\rho$ as evaluation metrics. The detailed description of evaluation metrics is provided in Appendix F.2.

### 4.2 QUANTITATIVE ANALYSIS

In this subsection, we present quantitative comparisons with state-of-the-art methods and conduct ablation studies on the frequency module to validate its effectiveness. More experiments and visualizations are provided in Appendix G.

**Intra-dataset Evaluation.** We present the HR evaluation results of our method compared to several representative baselines on four benchmarks in Tab. 1, adhering to the evaluation protocols established in prior work (Zou et al., 2025b; Qian et al., 2025). From the table, we observe that our method sets a new state-of-the-art performance on the more challenging MMPD and VIPL-HR datasets by a large margin. These findings demonstrate that incorporating physiological frequency priors enables our approach to effectively mitigate noise interference. In addition to HR estimation,

Table 3: Cross-dataset HR estimation results of the models trained on PURE/UBFC-rPPG and tested on UBFC-rPPG/PURE/MMPD.

| Method | Venue | PURE → UBFC-rPPG | | | UBFC-rPPG → PURE | | | PURE → MMPD | | | UBFC-rPPG → MMPD | | |
|---|---|---|---|---|---|---|---|---|---|---|---|---|---|
| | | MAE↓ | RMSE↓ | ρ↑ | MAE↓ | RMSE↓ | ρ↑ | MAE↓ | RMSE↓ | ρ↑ | MAE↓ | RMSE↓ | ρ↑ |
| DeepPhys (Chen & McDuff, 2018) | ECCV'18 | 1.21 | 2.90 | 0.99 | 5.54 | 18.51 | 0.66 | 16.92 | 24.61 | 0.05 | 17.50 | 25.00 | 0.05 |
| PhysNet (Yu et al., 2019) | BMVC'19 | 1.63 | 3.79 | 0.98 | 9.36 | 20.63 | 0.62 | 13.22 | 19.61 | 0.23 | 10.24 | 16.54 | 0.29 |
| TS-CAN (Liu et al., 2020) | NeurIPS'20 | 1.30 | 2.87 | 0.99 | 3.69 | 13.80 | 0.82 | 13.94 | 21.61 | 0.20 | 14.01 | 21.04 | 0.24 |
| PhysFormer (Yu et al., 2022) | CVPR'22 | 1.44 | 3.77 | 0.98 | 12.92 | 24.36 | 0.47 | 14.57 | 20.71 | 0.15 | 12.10 | 17.79 | 0.17 |
| EfficientPhys (Liu et al., 2023) | WACV'23 | 2.13 | 3.00 | 0.99 | 5.47 | 17.04 | 0.71 | 14.03 | 21.62 | 0.17 | 13.78 | 22.25 | 0.09 |
| RhythmMamba (Zou et al., 2025b) | AAAI'25 | 0.95 | 1.83 | 0.99 | 1.98 | 6.51 | 0.96 | 10.44 | 16.70 | 0.36 | 10.63 | 17.14 | 0.34 |
| PhysDiff (Qian et al., 2025) | AAAI'25 | 0.52 | 0.84 | **1.00** | 3.30 | 6.89 | 0.96 | 10.96 | 14.93 | 0.28 | 10.76 | 14.47 | 0.34 |
| **FrePhys (Ours)** | - | **0.43** | **0.79** | **1.00** | **0.95** | **3.15** | **0.99** | **10.11** | **14.34** | **0.48** | **8.91** | **12.86** | **0.57** |

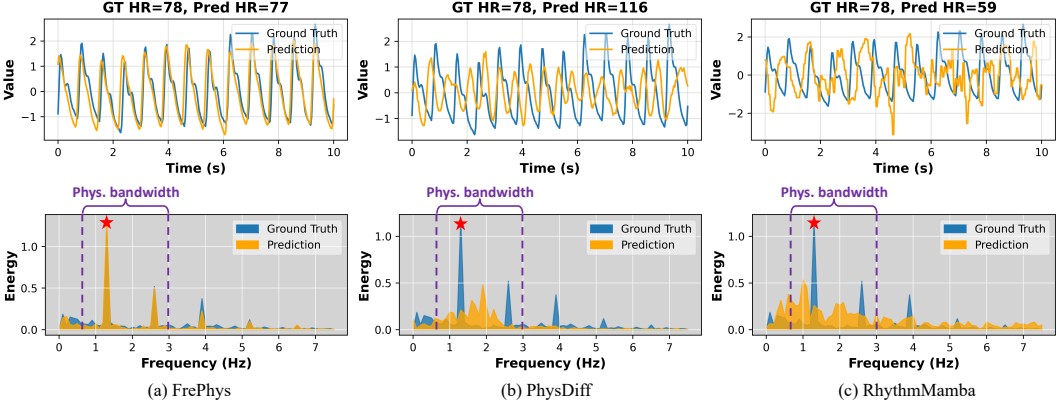

(a) FrePhys        (b) PhysDiff        (c) RhythmMamba

Figure 3: Time and frequency domain visualizations of rPPG signal predictions on the VIPL dataset under head motion scenario. The results are presented for (a) the proposed method, (b) PhysDiff (Qian et al., 2025), and (c) RhythmMamba (Zou et al., 2025b). In the frequency-domain plots, the purple dashed box indicates the physiological signal bandwidth ranging from 0.66 to 3.0 Hz, corresponding to typical human cardiac frequencies. ⋆ represents the spectrum peak of HR.

we also evaluate our method on two other critical physiological indicators, i.e., heart rate variability (HRV) and respiration frequency (RF), which require high-quality rPPG signals for accurate peak detection and reliable analysis. As shown in Tab. 2, our method significantly outperforms existing methods across most metrics. This demonstrates that our method not only captures precise cardiac pulsation cycles but also reconstructs high-fidelity rPPG signals in the time domain.

**Cross-dataset Evaluation.** As shown in Tab. 3, we conduct four cross-dataset evaluations to simulate unseen real-world scenarios. The results show that the performance of most models drops significantly when transferred from a simple domain to a complex domain, which is a challenge in this field. Benefiting from physiological spectrum modeling, our method effectively improves generalization and achieves the best performance in all settings.

**Ablation Studies.** We investigate the impact of different components in our method through the following ablation studies. As shown in Tab. 4, when physiological frequency information is missing, that is, only using time-domain MSTmap input as a condition, the performance degrades significantly. Additionally, it is evident that a series of physiological frequency denoiser modules play a crucial role, which verifies the necessity of each component within our method. The combination of PBF, PSM, and ASS achieves the best performance, highlighting that its combination brings unique benefits.

Table 4: Ablation study of the individual contributions on VIPL-HR.

| PBF | PSM | ASS | MAE | RMSE | ρ |
|---|---|---|---|---|---|
| - | - | - | 4.22 | 7.15 | 0.83 |
| ✓ | - | - | 4.03 | 6.77 | 0.82 |
| - | ✓ | - | 3.98 | 6.55 | 0.84 |
| - | - | ✓ | 4.10 | 6.95 | 0.82 |
| ✓ | ✓ | - | 3.91 | 6.42 | 0.85 |
| ✓ | - | ✓ | 4.11 | 6.71 | 0.84 |
| - | ✓ | ✓ | 3.95 | 6.48 | 0.83 |
| ✓ | ✓ | ✓ | **3.79** | **6.34** | **0.86** |

## 4.3 QUALITATIVE ANALYSIS

**Visualization of rPPG Prediction.** We visualize the rPPG predictions to highlight the improvements of our method in rPPG estimation quality. We present a prediction showcase on the VIPL dataset under the head motion scenario, as shown in Fig. 3. PhysDiff (Qian et al., 2025), Rhyth-

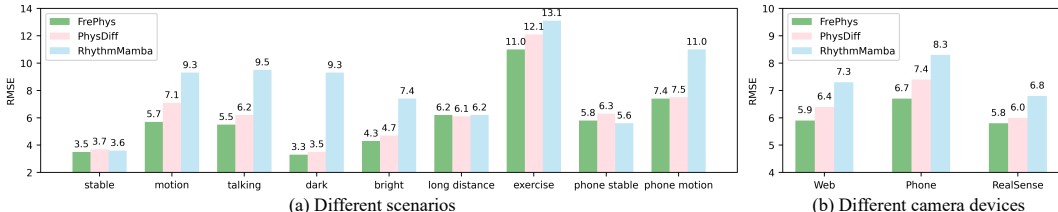

Figure 4: HR estimation results on VIPL-HR under different scenarios and camera devices.

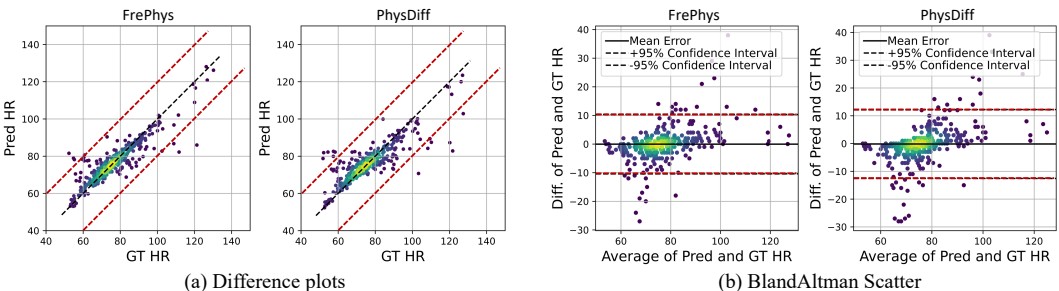

Figure 5: Difference plots and BlandAltman Scatter on VIPL-HR.

mMamba (Zou et al., 2025b) are selected as the representative SOTA methods. We observe that the rPPG signal predicted by PhysDiff and RhythmMamba exhibits numerous burrs, which reflect the limitations of modeling in the time domain, namely, the difficulty in capturing dominant physiological frequency components and being vulnerable to noise. Our FrePhys addresses this limitation effectively, which not only keeps pace with the label sequence but also accurately exhibits a smoother appearance with fewer irregularities. In addition, we also observe that the frequency spectra of PhysDiff and RhythmMamba exhibit relatively dispersed energy distributions. In contrast, our method effectively concentrates spectral energy on several key frequency components corresponding to physiological signals, resulting in an obvious sparsity in the frequency domain. This focused energy distribution highlights our method's ability to isolate and enhance vital physiological components while suppressing irrelevant noise.

**Qualitative Results for Robustness.** To evaluate the robustness of our model across diverse scenarios and camera device conditions, we provide detailed results from the VIPL-HR dataset, encompassing 9 distinct scenarios and 3 types of camera devices. As illustrated in Fig. 4, our proposed method consistently outperforms other methods across these varied conditions, which indicates its robustness in remote physiological signal measurement.

**Qualitative Results for Consistency.** To further evaluate the consistency between predicted HR and ground truth measurements across different ranges, Figure 5 presents both scatter and BlandAltman plots on the VIPL dataset. Compared to the diffusion-based PhysDiff (Qian et al., 2025), our proposed method shows scatter points more closely aligned with the identity line ($y = x$), indicating fewer outliers. Additionally, the BlandAltman plot reveals that our method exhibits narrower confidence intervals, suggesting reduced variability between predicted and actual HR values. These visualizations collectively indicate that our method achieves superior consistency with ground truth HR, highlighting its accuracy across all HR distributions.

## 5 CONCLUSION

In this paper, we introduced **FrePhys**, a novel frequency-aware diffusion model designed to enhance remote physiological estimation by integrating physiological frequency priors. Addressing the limitations of existing time-domain approaches, particularly their susceptibility to noise from motion artifacts and illumination variations, we leverage frequency-domain insights to improve signal fidelity. Extensive evaluations on multiple public datasets demonstrate that our method outperforms state-of-the-art methods in HR, HRV, and RF estimation tasks. Notably, our method exhibits superior generalization capabilities in cross-dataset scenarios, underscoring its robustness in diverse and challenging conditions.

## 6 ETHICS STATEMENT

This work complies with the ICLR Code of Ethics. No human or animal experiments were conducted. All datasets used (UBFC-rPPG, PURE, MMPD, and VIPL-HR) were obtained in accordance with their respective usage policies and do not contain personally identifiable information. We ensured that our methods do not raise privacy, security, or fairness concerns, and we took care to minimize potential biases. We are committed to maintaining transparency, integrity, and ethical responsibility throughout this research.

## 7 REPRODUCIBILITY STATEMENT

We have taken extensive steps to ensure the reproducibility of our results. An anonymous repository provides full access to our code and processed datasets. Detailed descriptions of the experimental setup, including training procedures, model configurations, and hardware specifications, are included in the paper. These measures are intended to facilitate replication and foster further research in this area.

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

# Appendix for ❤️FrePhys: Frequency-aware Diffusion Model for Remote Physiological Measurement

| | |
|---|---|
| Section A | **Detailed Related Work** |
| Section B | **Preliminaries** |
| Section C | **Theoretical Proof** |
| Section D | **Mathematical Derivation Details** |
| Section E | **Model Details** |
| Section F | **Reproduction Details** |
| Section G | **Additional Experimental Results** |
| Section H | **Limitations ans Future Work** |
| Section I | **LLM Usage** |

## A  DETAILED RELATED WORK

### A.1  REMOTE PHYSIOLOGICAL MEASUREMENT

The physiological mechanism of rPPG lies in the periodic changes in subcutaneous blood volume driven by cardiac contraction and relaxation. These fluctuations alter the skin's absorption and scattering of light, producing subtle color variations that are imperceptible to the human eye. Early work focused on hand-crafted methods, including blind source separation (BSS) approaches (Poh et al., 2010; Lewandowska et al., 2011) using PCA/ICA to isolate rPPG signals from noise, and model-driven techniques like CHROM and POS (De Haan & Jeanne, 2013; Li et al., 2014), which leverage color space projections based on physiological priors. These methods perform well under constrained settings but degrade significantly with motion or illumination changes. Deep learning methods then began to flourish. HR-CNN (Špetlík et al., 2018) is the first work using deep learning models for rPPG, which proposed a two-step convolutional neural network to estimate HR value. DeepPhys (Chen & McDuff, 2018) then proposed to estimate BVP signals from the normalized difference of adjacent frames and to use raw facial images to adaptively generate attention maps to guide the estimation. CVD (Niu et al., 2020) utilized the multi-scale spatial-temporal map to represent physiological features in raw facial videos and proposed the cross-verified feature disentangling strategy to separate noise features and physiological features. Dual-GAN (Lu et al., 2021) employed two GANs to jointly model BVP prediction and noise distribution to improve robustness across facial regions. To capture long-range temporal dependencies, subsequent research (PhysFormer (Yu et al., 2022), EfficientPhys (Liu et al., 2023), Dual-TL (Qian et al., 2024a), RhythmFormer (Zou et al., 2025a)) turned to the Transformer architecture. To further maintain linear complexity, some Mamba-based work (RhythmMamba (Zou et al., 2025b), PhysMamba (Luo et al., 2024)) then introduced the state space model, achieving high performance with low memory usage and improved speed. Due to the scarcity of labeled data, self-supervised learning has gained attention. Contrastive approaches (Gideon & Stent, 2021; Sun & Li, 2024) and masked autoencoding (Liu et al., 2024; Speth et al., 2023) enable robust representation learning from unlabeled videos, showing strong potential for in-the-wild applications.

### A.2  DIFFUSION MODEL FOR RPPG ESTIMATION

Diffusion models have emerged as powerful generative frameworks that gradually corrupt training data with noise and learn to reverse this process to generate clean samples. Initially proposed for image generation (Ho et al., 2020), denoising diffusion probabilistic models (DDPMs) have since been successfully applied to a broad range of domains, including cross-modal generation (Avrahami et al., 2022; Fan et al., 2022), video editing (Ceylan et al., 2023), and object detection (Chen et al., 2022). In remote physiological measurement, particularly rPPG estimation, the challenges of complex motion, illumination variability, and weak signal strength motivate the need for robust

denoising techniques. Diffusion models, with their capacity to model complex data distributions and restore clean signals, offer a promising direction. A recent pioneering work, PhysDiff (Qian et al., 2025), introduces diffusion to the rPPG field by designing a dynamic-aware signal representation. PhysDiff decomposes rPPG signals into two components: trend, representing temporal directionality (capillary expansion/contraction), and amplitude, quantifying signal fluctuation intensity. Building upon this direction, our proposed method explores a frequency-aware perspective for rPPG estimation using diffusion models. Inspired by the frequency modeling being beneficial for sequence signal analysis (Yi et al., 2023; 2024), we try to introduce the physiological frequency prior into rPPG estimation. While PhysDiff emphasizes time-domain signal dynamics, our method shifts focus to the periodic nature of rPPG by modeling frequency-domain features.

## B PRELIMINARIES

### B.1 MULTIPLICATION OF COMPLEX NUMBERS

Consider two complex numbers $\mathcal{Z}_1 = a + jb$ and $\mathcal{Z}_2 = c + jd$, where $a$ and $c$ denote the real parts of $\mathcal{Z}_1$ and $\mathcal{Z}_2$, respectively, and $b$ and $d$ represent the corresponding imaginary parts. The multiplication of two complex numbers involves applying the distributive property of multiplication over addition, along with the identity $j^2 = -1$, where $j$ is the imaginary unit. The product of $\mathcal{Z}_1$ and $\mathcal{Z}_2$ is computed as follows:

$$
\begin{aligned}
\mathcal{Z}_1 \mathcal{Z}_2 &= (a + jb)(c + jd) \\
&= ac + a(jd) + jb(c) + jb(jd) \\
&= ac + j(ad) + j(bc) + j^2(bd) \\
&= (ac - bd) + j(ad + bc).
\end{aligned}
\tag{18}
$$

### B.2 DISCRETE FOURIER TRANSFORM

The *Discrete Fourier Transform* (DFT) (Brigham & Morrow, 1967) is a fundamental tool in signal processing and spectral analysis. It transforms a discrete-time signal from the temporal domain to the frequency domain, enabling a decomposition of the signal into its constituent frequency components. This facilitates the precise identification and analysis of underlying periodic patterns and oscillatory behavior.

Given a discrete real-valued temporal signal $\boldsymbol{x} \in \mathbb{R}^T$, its frequency-domain representation $\boldsymbol{x}^F \in \mathbb{C}^T$ is a complex-valued sequence defined by:

$$
\boldsymbol{x}^F[i] = \sum_{t=0}^{T-1} \boldsymbol{x}[t] \cdot e^{-j2\pi it/T} = \underbrace{\sum_{t=0}^{T-1} \boldsymbol{x}[t] \cdot \cos\left(\frac{2\pi it}{T}\right)}_{\text{Real Part}} - j\underbrace{\sum_{t=0}^{T-1} \boldsymbol{x}[t] \cdot \sin\left(\frac{2\pi it}{T}\right)}_{\text{Imaginary Part}},
\tag{19}
$$

where $i \in 0, 1, \ldots, T-1$ indexes the discrete frequency bins, and $j$ is the imaginary unit such that $j^2 = -1$. The corresponding physical frequency for the $i$-th bin is given by $\lambda_i = if_s/T$ Hz, where $f_s$ is the sampling rate of the signal $\boldsymbol{x}$. For real-valued signals, the DFT exhibits conjugate symmetry:

$$
\boldsymbol{x}^F[T - n] = \overline{\boldsymbol{x}^F[n]}, \quad \text{for } n = 1, \ldots, \lfloor T/2 \rfloor,
\tag{20}
$$

allowing us to retain only the first $\lfloor T/2 \rfloor + 1$ frequency components without loss of information. Hence, in practice, we define the DFT operator as a mapping $\mathcal{F} : \mathbb{R}^T \to \mathbb{C}^{\lfloor T/2 \rfloor + 1}$ for computational efficiency. Each complex coefficient $\boldsymbol{x}^F[i]$ in the frequency domain can be uniquely expressed in terms of its amplitude and phase:

$$
A[i] = |\boldsymbol{x}^F[i]| = \sqrt{\text{Re}(\boldsymbol{x}^F[i])^2 + \text{Im}(\boldsymbol{x}^F[i])^2}, \qquad \phi[i] = \tan^{-1}\left(\frac{\text{Im}(\boldsymbol{x}^F[i])}{\text{Re}(\boldsymbol{x}^F[i])}\right),
\tag{21}
$$

where $\text{Re}(\cdot)$ and $\text{Im}(\cdot)$ denote the real and imaginary parts, respectively. The amplitude $A[i]$ reflects the energy concentration at frequency $\lambda_i$, while the phase $\phi[i]$ captures the temporal alignment of the sinusoidal component at that frequency.

Since the DFT is a bijective (invertible) transformation, the original time-domain signal $\boldsymbol{x}[t]$ can be exactly reconstructed from its frequency-domain representation $\boldsymbol{x}^F[i]$ via the Inverse Discrete Fourier Transform (IDFT), expressed as:

$$\boldsymbol{x}[t] = \mathcal{F}^{-1}(\boldsymbol{x}^F)[t] = \frac{1}{T} \sum_{i=0}^{T-1} \boldsymbol{x}^F[i] \cdot e^{j2\pi it/T}, \quad t = 0, 1, \ldots, T-1. \tag{22}$$

# C  THEORETICAL PROOF

## C.1  PROOF OF THEOREM 1

**Theorem 1** *(Frequency-domain Convolution Theorem) The multiplication of two signals in the frequency domain is equivalent to the frequency transformation of a circular convolution of these two signals in the temporal domain, which can be summarized as:*

$$\mathcal{F}[\mathbf{M}(v) \otimes \mathbf{Z}(v)] = \mathcal{F}(\mathbf{M}(v)) \odot \mathcal{F}(\mathbf{Z}(v)), \tag{23}$$

*where $\otimes$ and $\odot$ represent the circular convolutional operation and element-multiplication operation, respectively, $\mathbf{M}(v)$ and $\mathbf{Z}(v)$ represent two signals for the time variable $v$, and $\mathcal{F}(\cdot)$ denotes the Discrete Fourier Transform.*

*Proof.* Let $\mathbf{M}(v)$ and $\mathbf{Z}(v)$ are two length $T$ signals. Let $\mathbf{M}(v)$ and $\mathbf{Z}(v)$ be two discrete signals of length $T$, defined over the time index $v = 0, 1, \ldots, T-1$. Let their DFTs be denoted by $\mathcal{F}(\mathbf{M}(v))$ and $\mathcal{F}(\mathbf{Z}(v))$, respectively. We define the circular convolution of $\mathbf{M}(v)$ and $\mathbf{Z}(v)$ as:

$$\mathbf{Y}(v) = \mathbf{M}(v) \otimes \mathbf{Z}(v) = \sum_{u=0}^{T-1} \mathbf{M}(u) \cdot \mathbf{Z}((v-u) \mod T). \tag{24}$$

The DFT of the resulting signal $\mathbf{Y}(v)$ is given by:

$$\mathcal{F}(\mathbf{Y}(v)) = \sum_{v=0}^{T-1} \mathbf{Y}(v) \cdot e^{-j2\pi iv/T}, \quad i = 0, 1, \ldots, T-1, \tag{25}$$

where $j$ is the imaginary unit, and $i$ denotes the frequency index. Substituting the expression for $\mathbf{Y}(v)$ into the DFT, we obtain:

$$\begin{aligned} \mathcal{F}(\mathbf{Y}(v)) &= \sum_{v=0}^{T-1} \left( \sum_{u=0}^{T-1} \mathbf{M}(u) \cdot \mathbf{Z}((v-u) \mod T) \right) e^{-j2\pi iv/T} \\ &= \sum_{u=0}^{T-1} \mathbf{M}(u) \cdot \sum_{v=0}^{T-1} \mathbf{Z}((v-u) \mod T) \cdot e^{-j2\pi iv/T}. \end{aligned} \tag{26}$$

Next, we perform a change of variable by letting $r = (v - u) \mod T$, which implies $v = (r + u) \mod T$:

$$\begin{aligned} \mathcal{F}(\mathbf{Y}(v)) &= \sum_{u=0}^{T-1} \mathbf{M}(u) \cdot \sum_{r=0}^{T-1} \mathbf{Z}(r) \cdot e^{-j2\pi i(r+u)/T} \\ &= \sum_{u=0}^{T-1} \mathbf{M}(u) \cdot e^{-j2\pi iu/T} \cdot \sum_{r=0}^{T-1} \mathbf{Z}(r) \cdot e^{-j2\pi ir/T}. \end{aligned} \tag{27}$$

Rewriting the above expression, we have:

$$\mathcal{F}(\mathbf{Y}(v)) = \left( \sum_{u=0}^{T-1} \mathbf{M}(u) \cdot e^{-j2\pi iu/T} \right) \cdot \left( \sum_{r=0}^{T-1} \mathbf{Z}(r) \cdot e^{-j2\pi ir/T} \right) = \mathcal{F}(\mathbf{M}(v)) \odot \mathcal{F}(\mathbf{Z}(v)). \tag{28}$$

Thus, the Discrete Fourier Transform of the circular convolution of two signals $\mathbf{M}(v)$ and $\mathbf{Z}(v)$ is equivalent to the element-wise product of their respective DFTs, i.e., $\mathcal{F}[\mathbf{M}(v) \otimes \mathbf{Z}(v)] = \mathcal{F}(\mathbf{M}(v)) \odot \mathcal{F}(\mathbf{Z}(v))$. This completes the proof.

### C.2 PROOF OF PROPOSITION 1

**Proposition 1** *The conditional entropy is satisfied:*

$$\mathbf{H}(\mathbf{Y}_{k-1}|\mathbf{Y}_k, \mathbf{X}, \mathbf{C}^{\mathbf{P}}) < \mathbf{H}(\mathbf{Y}_{k-1}|\mathbf{Y}_k, \mathbf{X}), \tag{29}$$

*indicating that the inclusion of additional physiological frequency condition $\mathbf{C}^{\mathbf{P}}$ in the reverse process reduces uncertainty.*

*Proof.* We use the notion of conditional entropy from information theory to quantify uncertainty. In the reverse process of DDPM (Ho et al., 2020), the rPPG signal at step $k$, denoted $\mathbf{Y}_k$, is treated as a condition. The uncertainty of the reverse process can thus be expressed as:

$$\mathbf{H}\left(\mathbf{Y}_{k-1} \mid \mathbf{Y}_k\right) = -\int p_\theta\left(\mathbf{Y}_{k-1}, \mathbf{Y}_k\right) \log p_\theta\left(\mathbf{Y}_{k-1} \mid \mathbf{Y}_k\right) \mathrm{d}\mathbf{Y}_{k-1}. \tag{30}$$

Similarly, in PhysDiff (Qian et al., 2025), the condition includes only the facial observation sequence $\mathbf{X}$, so the uncertainty is modeled as $\mathbf{H}(\mathbf{Y}_{k-1} \mid \mathbf{Y}_k, \mathbf{X})$. In our proposed method, the condition is extended to include physiological frequency information $\mathbf{C}^{\mathbf{P}}$, yielding $\mathbf{H}(\mathbf{Y}_{k-1} \mid \mathbf{Y}_k, \mathbf{X}, \mathbf{C}^{\mathbf{P}})$.

From the basic property of conditional entropy, we know:

$$\mathbf{H}(\mathbf{Y}_{k-1} \mid \mathbf{Y}_k) \leq \mathbf{H}(\mathbf{Y}_{k-1}). \tag{31}$$

Using the definition of mutual information, we have:

$$\mathbf{I}(\mathbf{Y}_{k-1}; \mathbf{Y}_k) = \mathbf{H}(\mathbf{Y}_{k-1}) - \mathbf{H}(\mathbf{Y}_{k-1} \mid \mathbf{Y}_k). \tag{32}$$

According to Equation 11, we know that $\mathbf{I}(\mathbf{Y}_{k-1}; \mathbf{Y}_k) > 0$, which implies:

$$\mathbf{H}(\mathbf{Y}_{k-1} \mid \mathbf{Y}_k) < \mathbf{H}(\mathbf{Y}_{k-1}). \tag{33}$$

Using the chain rule for entropy, we have:

$$\mathbf{H}(\mathbf{Y}_{k-1}, \mathbf{Y}_k, \mathbf{X}) = \mathbf{H}(\mathbf{Y}_{k-1} \mid \mathbf{Y}_k, \mathbf{X}) + \mathbf{H}(\mathbf{Y}_k, \mathbf{X}). \tag{34}$$

Rewriting this using conditional entropy identities:

$$\begin{aligned} \mathbf{H}(\mathbf{Y}_{k-1} \mid \mathbf{Y}_k, \mathbf{X}) &= \mathbf{H}(\mathbf{X} \mid \mathbf{Y}_{k-1}, \mathbf{Y}_k) + \mathbf{H}(\mathbf{Y}_{k-1}, \mathbf{Y}_k) - \mathbf{H}(\mathbf{X} \mid \mathbf{Y}_k) - \mathbf{H}(\mathbf{Y}_k) \\ &= \mathbf{H}(\mathbf{Y}_{k-1} \mid \mathbf{Y}_k) + \mathbf{H}(\mathbf{X} \mid \mathbf{Y}_{k-1}, \mathbf{Y}_k) - \mathbf{H}(\mathbf{X} \mid \mathbf{Y}_k). \end{aligned} \tag{35}$$

From Equation 11, since $\mathbf{Y}_{k-1}$ contains one less step of noise compared to $\mathbf{Y}_k$, it is closer to the original observation. Therefore:

$$\mathbf{H}(\mathbf{X} \mid \mathbf{Y}_{k-1}, \mathbf{Y}_k) < \mathbf{H}(\mathbf{X} \mid \mathbf{Y}_k). \tag{36}$$

Substituting this inequality back gives:

$$\mathbf{H}(\mathbf{Y}_{k-1} \mid \mathbf{Y}_k, \mathbf{X}) < \mathbf{H}(\mathbf{Y}_{k-1} \mid \mathbf{Y}_k). \tag{37}$$

Following the same reasoning, we can conclude:

$$\mathbf{H}(\mathbf{Y}_{k-1} \mid \mathbf{Y}_k, \mathbf{X}, \mathbf{C}^{\mathbf{P}}) < \mathbf{H}(\mathbf{Y}_{k-1} \mid \mathbf{Y}_k, \mathbf{X}). \tag{38}$$

This result confirms that incorporating the physiological prior $\mathbf{C}^{\mathbf{P}}$ into the conditioning set reduces the entropy of the target distribution in the reverse process. Consequently, this reduction in uncertainty simplifies the learning task for the diffusion model, potentially leading to more efficient training and enhanced accuracy in rPPG signal estimation. This completes the proof.

## D MATHEMATICAL DERIVATION DETAILS

### D.1 DERIVATIONS OF CLOSED-FORM FORWARD PROCESS

Assuming that the clean rPPG target distribution $q(\mathbf{Y}_0)$ is known, we can first sample a clean rPPG target as $\mathbf{Y}_0 \sim q(\mathbf{Y}_0)$. According to the forward process defined in Equation 11, the noisy rPPG signal at step $k$ can be generated as:

$$\mathbf{Y}_k = \sqrt{\alpha_k}\mathbf{Y}_{k-1} + \sqrt{1 - \alpha_k}\epsilon_k, \quad \epsilon_k \sim \mathcal{N}(\mathbf{0}, \mathbf{I}). \tag{39}$$

Similarly, the previous step $\mathbf{Y}{k-1}$ can be expressed as:

$$\mathbf{Y}_{k-1} = \sqrt{\alpha_{k-1}}\mathbf{Y}_{k-2} + \sqrt{1 - \alpha_{k-1}}\epsilon_{k-1}, \quad \epsilon_{k-1} \sim \mathcal{N}(\mathbf{0}, \mathbf{I}). \tag{40}$$

By recursively substituting, we obtain:

$$\begin{aligned}
\mathbf{Y}_k &= \sqrt{\alpha_k}\left(\sqrt{\alpha_{k-1}}\mathbf{Y}_{k-2} + \sqrt{1 - \alpha_{k-1}}\epsilon_{k-1}\right) + \sqrt{1 - \alpha_k}\epsilon_k \\
&= \sqrt{\alpha_k\alpha_{k-1}}\mathbf{Y}_{k-2} + \left(\sqrt{\alpha_k(1 - \alpha_{k-1})}\epsilon_{k-1} + \sqrt{1 - \alpha_k}\epsilon_k\right).
\end{aligned} \tag{41}$$

Given that $\epsilon_{k-1}, \epsilon_k \sim \mathcal{N}(\mathbf{0}, \mathbf{I})$, the two noise terms are independent Gaussian variables. Therefore, their weighted sum is also Gaussian:

$$\begin{aligned}
\sqrt{\alpha_k(1 - \alpha_{k-1})}\epsilon_{k-1} &\sim \mathcal{N}(\mathbf{0}, \alpha_k(1 - \alpha_{k-1})\mathbf{I}), \\
\sqrt{1 - \alpha_k}\epsilon_k &\sim \mathcal{N}(\mathbf{0}, (1 - \alpha_k)\mathbf{I}),
\end{aligned} \tag{42}$$

and their sum follows:

$$\sqrt{\alpha_k(1 - \alpha_{k-1})}\epsilon_{k-1} + \sqrt{1 - \alpha_k}\epsilon_k \sim \mathcal{N}\left(\mathbf{0}, [\alpha_k(1 - \alpha_{k-1}) + (1 - \alpha_k)]\mathbf{I}\right). \tag{43}$$

This implies that the overall expression can be rewritten in the same form as before:

$$\mathbf{Y}_k = \sqrt{\alpha_k\alpha_{k-1}}\mathbf{Y}_{k-2} + \sqrt{1 - \alpha_k\alpha_{k-1}}\epsilon, \tag{44}$$

where $\epsilon \sim \mathcal{N}(\mathbf{0}, \mathbf{I})$.

Continuing this recursive process, we eventually obtain:

$$\mathbf{Y}k = \sqrt{\prod_{s=1}^{k}\alpha_s}\mathbf{Y}_0 + \sqrt{1 - \prod_{s=1}^{k}\alpha_s}\epsilon, \quad \epsilon \sim \mathcal{N}(\mathbf{0}, \mathbf{I}). \tag{45}$$

This result corresponds to the closed-form expression of $\mathbf{Y}_k$ in the forward diffusion process, starting from a clean rPPG signal $\mathbf{Y}_0$.

### D.2 DERIVATION OF PARAMETERIZED REVERSE PROCESS

We begin with Bayes' theorem to derive the reverse process:

$$p_\theta(\mathbf{Y}_{k-1}|\mathbf{Y}_k, \mathbf{X}, \mathbf{C^P}) = p_\theta(\mathbf{Y}_k|\mathbf{Y}_{k-1}, \mathbf{X}, \mathbf{C^P})\frac{p_\theta(\mathbf{Y}_{k-1}|\mathbf{X}, \mathbf{C^P})}{p_\theta(\mathbf{Y}_k|\mathbf{X}, \mathbf{C^P})} \tag{46}$$

According to Equation 11, the expected $p_\theta(\mathbf{Y}_k|\mathbf{Y}_{k-1}, \mathbf{X}, \mathbf{C^P})$ is:

$$p_\theta(\mathbf{Y}_k|\mathbf{Y}_{k-1}, \mathbf{X}, \mathbf{C^P}) \sim \mathcal{N}(\mathbf{Y}_k; \sqrt{\alpha_k}\mathbf{Y}_{k-1}, \beta_k\mathbf{I}). \tag{47}$$

Furthermore, based on Equation 13, we also have:

$$\begin{aligned}
p_\theta(\mathbf{Y}_{k-1}|\mathbf{X}, \mathbf{C^P}) &\sim \mathcal{N}(\mathbf{Y}_{k-1}; \sqrt{\bar{\alpha}_{k-1}}\mathbf{Y}_0, (1 - \bar{\alpha}_{k-1})\mathbf{I}), \\
p_\theta(\mathbf{Y}_k|\mathbf{X}, \mathbf{C^P}) &\sim \mathcal{N}(\mathbf{Y}_{k-1}; \sqrt{\bar{\alpha}_k}\mathbf{Y}_0, (1 - \bar{\alpha}_k)\mathbf{I}),
\end{aligned} \tag{48}$$

Combining the above three Gaussian distributions, we can derive:

$$p_\theta(\mathbf{Y}_{k-1}|\mathbf{Y}_k, \mathbf{X}, \mathbf{C^P}) \propto \mathcal{N}(\mathbf{Y}_k; \sqrt{\alpha_k}\mathbf{Y}_{k-1}, (1 - \alpha_k) \cdot \mathcal{N}(\mathbf{Y}_{k-1}; \sqrt{\bar{\alpha}_{k-1}}\mathbf{Y}_0, (1 - \bar{\alpha}_{k-1})\mathbf{I}) \tag{49}$$

Since the product of two Gaussians is also a Gaussian, we can compute the posterior distribution analytically using the standard Gaussian product rule. Specifically, the reverse process becomes:

$$p_\theta(\mathbf{Y}_{k-1}|\mathbf{Y}_k, \mathbf{X}, \mathbf{C^P}) = \mathcal{N}(\mathbf{Y}_{k-1}; \mu_\theta(\mathbf{Y}_k, \mathbf{X}, \mathbf{C^P}, k), \Sigma_\theta). \tag{50}$$

where the mean and covariance are given by:

$$\begin{aligned}
\Sigma_\theta &= \left(\frac{1}{\beta_k}\mathbf{I} + \frac{1}{1 - \bar{\alpha}_{k-1}}\mathbf{I}\right)^{-1} = \frac{(1 - \bar{\alpha}_{k-1})\beta_k}{1 - \bar{\alpha}_k}\mathbf{I}, \\
\mu_\theta(\mathbf{Y}_k, \mathbf{X}, \mathbf{C^P}, k) &= \Sigma_\theta\left(\frac{1}{\beta_k}\sqrt{\alpha_k}\mathbf{Y}_k + \frac{1}{1 - \bar{\alpha}_{k-1}}\sqrt{\bar{\alpha}_{k-1}}\mathbf{Y}_0\right) = \frac{\sqrt{\alpha_k}(1 - \bar{\alpha}_{k-1})}{1 - \bar{\alpha}_k}\mathbf{Y}_k + \frac{\sqrt{\bar{\alpha}_{k-1}}\beta_k}{1 - \bar{\alpha}_k}\mathbf{Y}_0.
\end{aligned} \tag{51}$$

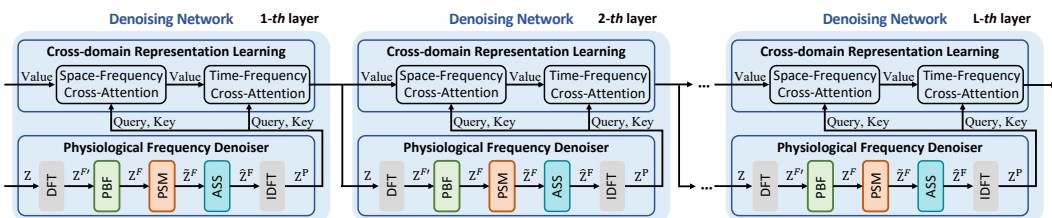

Figure 6: Implementation details of the stacked denoising network in our **FrePhys**.

However, during inference, the clean signal $\mathbf{Y}_0$ is not directly accessible. Therefore, we train a neural network $f_\theta(\cdot)$ to predict an approximation $\hat{\mathbf{Y}}_0$ from the noisy input:

$$\hat{\mathbf{Y}}_0 = f_\theta(\mathbf{Y}_k, \mathbf{X}, \mathbf{C^P}, k), \tag{52}$$

and substitute $\mathbf{Y}_0$ in the mean computation $\mu_\theta(\mathbf{Y}_k, \mathbf{X}, \mathbf{C^P}, k)$ with the predicted $\hat{\mathbf{Y}}_0$. This yields the parameterized reverse process:

$$\mu_\theta(\mathbf{Y}_k, \mathbf{X}, \mathbf{C^P}, k) = \frac{\sqrt{\alpha_k}(1 - \bar{\alpha}_{k-1})}{1 - \bar{\alpha}_k} \mathbf{Y}_k + \frac{\sqrt{\bar{\alpha}_{k-1}}\beta_k}{1 - \bar{\alpha}_k} \hat{\mathbf{Y}}_0, \tag{53}$$

where $\hat{\mathbf{Y}}_0 = f_\theta(\mathbf{Y}_k, \mathbf{X}, \mathbf{C^P}, k)$. The variance is kept fixed as:

$$\Sigma_\theta = \sigma_k^2 \mathbf{I}, \quad \text{with} \quad \sigma_k^2 = \frac{(1 - \bar{\alpha}_{k-1})\beta_k}{1 - \bar{\alpha}_k}. \tag{54}$$

Therefore, this approach enables efficient learning by directly predicting the clean rPPG signal $\mathbf{Y}_0$, thereby avoiding explicit noise estimation as in the original DDPM framework (Ho et al., 2020).

# E   MODEL DETAILS

## E.1   DETAILED ARCHITECTURE OF THE DENOISING NETWORK

As shown in Fig. 6, we provide the implementation details of the stacked denoising network..

## E.2   CROSS-DOMAIN REPRESENTATION LEARNING

**Space-Frequency Cross-Attention.** To capture spatial dependencies across facial ROIs guided by physiological frequency clues, we apply a multi-head cross-attention over the ROI dimension at each timestamp. Assuming the inputs of $l$-th layer are the intermediate feature $\mathbf{Z}^{(l)} \in \mathbb{R}^{T \times N \times D}$ and physiological frequency representation $\mathbf{Z}^{\mathbf{P},(l)} \in \mathbb{R}^{T \times N \times D}$. Then, for each timestamp $t$, the process of space-frequency interaction learning is formulated as:

$$Q_t = \mathbf{Z}_t^{\mathbf{P},(l)} W_S^Q, \quad K_t = \mathbf{Z}_t^{\mathbf{P},(l)} W_S^K, \quad V_t = \mathbf{Z}_t^{(l)} W_S^V,$$

$$\mathbf{Z}_t^{\mathbf{S},(l)} = \text{LayerNorm}\left(\text{Softmax}\left(\frac{Q_t K_t^\top}{\sqrt{D}}\right) V_t + \mathbf{Z}_t^{(l)}\right), \tag{55}$$

$$\hat{\mathbf{Z}}_t^{\mathbf{S},(l)} = \text{LayerNorm}\left(\mathbf{Z}_t^{\mathbf{S},(l)} + \text{FeedForward}\left(\mathbf{Z}_t^{\mathbf{S},(l)}\right)\right).$$

where $W_S^Q, W_S^K, W_S^V \in \mathbb{R}^{D \times D}$ are learnable projection matrices. Finally, the outputs $\hat{\mathbf{Z}}^{\mathbf{T},(l)}$ at all timestamps are concatenated along the temporal dimension to obtain the updated intermediate feature:

$$\mathbf{Z}^{\mathbf{S},(l)} \leftarrow \text{Concat}\left(\{\hat{\mathbf{Z}}_t^{\mathbf{S},(l)}\}_{t=1}^T\right). \tag{56}$$

**Time-Frequency Cross-Attention.** Complementary to space-frequency cross-attention, time-frequency cross-attention further models the temporal periodic dependencies within individual ROIs

through frequency-guided cross-attention along the time axis. For each ROI $n$, it can be formulated as:

$$Q_n = \mathbf{Z}_n^{\mathbf{P},(l)} W_T^Q, \quad K_n = \mathbf{Z}_n^{\mathbf{P},(l)} W_T^K, \quad V_n = \mathbf{Z}_n^{\mathbf{S},(l)} W_T^V,$$

$$\mathbf{Z}_n^{\mathbf{T},(l)} = \text{LayerNorm}\left(\text{Softmax}\left(\frac{Q_n K_n^\top}{\sqrt{D}}\right) V_n + \mathbf{Z}_n^{\mathbf{S},(l)}\right), \quad (57)$$

$$\hat{\mathbf{Z}}_n^{\mathbf{T},(l)} = \text{LayerNorm}\left(\mathbf{Z}_n^{\mathbf{T},(l)} + \text{FeedForward}\left(\mathbf{Z}_n^{\mathbf{T},(l)}\right)\right).$$

where $W_T^Q, W_T^K, W_T^V \in \mathbb{R}^{D \times D}$ are independent learnable projection matrices. Finally, the outputs $\hat{\mathbf{Z}}_t^{\mathbf{T},(l)}$ for all ROIs are concatenated along the spatial dimension to update the intermediate feature:

$$\mathbf{Z}^{(l+1)} \leftarrow \text{Concat}\left(\{\hat{\mathbf{Z}}_n^{\mathbf{T},(l)}\}_{n=1}^N\right). \quad (58)$$

# F  REPRODUCTION DETAILS

## F.1  DATASETS DETAILS

Our experiments for HR estimation are conducted on four publicly available datasets:

**UBFC-rPPG** (Bobbia et al., 2019) is a small-scale yet widely used dataset consisting of 42 facial videos from 42 subjects. Participants are recorded while performing time-limited mental arithmetic tasks, designed to introduce natural heart rate variability. The videos are of high quality with minimal noise or motion artifacts, making UBFC-rPPG an ideal benchmark for evaluating model accuracy under relatively clean and controlled conditions. According to the previous protocol (Lu et al., 2021; Song et al., 2021b; Qian et al., 2024a; Zou et al., 2025b), we select subjects 38 to 49 as the test set, and the remaining subjects are used as the training set.

**PURE** (Stricker et al., 2014) is another small-scale dataset designed for testing under controlled motion conditions. It comprises 60 one-minute videos from 10 subjects, each participating in six scenarios: 1) sitting still, 2) talking, 3) slow head movement, 4) fast head movement, 5) small head rotation, and 6) medium head rotation. This dataset introduces moderate motion artifacts and is suitable for evaluating model robustness to dynamic facial movements while maintaining good temporal synchronization. Following previous research (Sun & Li, 2022; Zou et al., 2025b; Qian et al., 2025), we divided the PURE dataset into a training set and a test set with a 6:4 ratio.

**MMPD** (Tang et al., 2023) is a medium-scale, challenging dataset featuring 660 one-minute videos from 33 subjects, each recorded under a wide range of conditions. MMPD is designed to simulate real-world complexity by including noise, motion, and illumination variation. In our study, we use the full uncompressed version of MMPD, making it a strong testbed for evaluating the generalization capabilities of rPPG methods under practical deployment scenarios. Following the protocols outlined in (Zou et al., 2025b;a), the dataset was sequentially split into training, validation, and testing sets with a ratio of 7:1:2.

**VIPL-HR** (Niu et al., 2019) is a large-scale rPPG dataset composed of 2,378 facial RGB videos from 107 subjects. Videos are recorded using three types of devices: Logitech C310 web-camera, the frontal camera of HUAWEI P9 phone, and RealSense F200 camera, under nine different scenarios combining varied illumination (e.g., bright, dark) and head motion (e.g., stable, talking, motion). Due to its diversity in capture devices, environments, and subject behaviors, VIPL-HR provides a robust benchmark for assessing the generalization and reliability of rPPG models in real-world conditions. Following previous work (Niu et al., 2019; 2020; Qian et al., 2024b; 2025), we use a subject-exclusive 5-fold cross-validation protocol on VIPL-HR in our experiments.

## F.2  EVALUATION METRICS

Following standard protocols in prior works (Li et al., 2014; Chen & McDuff, 2018; Niu et al., 2020), we adopt three commonly used metrics to evaluate the accuracy of heart rate (HR) estimation: *mean absolute error* (MAE), *root mean square error* (RMSE), and *Pearson's correlation coefficient* ($\rho$).

For the evaluation of heart rate variability (HRV) features, including respiration frequency (RF), low-frequency (LF) power in normalized units (n.u.), high-frequency (HF) power in normalized

Table 5: Ablation study of different physiological bandwidth.

| Physiological Bandwidth | MMPD | | | VIPL-HR | | |
|---|---|---|---|---|---|---|
| | MAE↓ | RMSE↓ | $\rho$↑ | MAE↓ | RMSE↓ | $\rho$↑ |
| $[0.75, 2.5]$ | 4.39 | 7.17 | 0.85 | 3.96 | 6.58 | 0.86 |
| $[0.75, 3.0]$ | 4.21 | 6.98 | 0.86 | 3.84 | 6.42 | 0.86 |
| $[0.66, 2.5]$ | 4.22 | 7.12 | 0.85 | 3.89 | 6.50 | 0.85 |
| $[0.66, 3.0]$ | **4.20** | **6.78** | **0.86** | **3.79** | **6.34** | **0.86** |

units (n.u.), and the LF/HF power ratio, we follow (Lu et al., 2021; Sun & Li, 2024), and utilize *standard deviation* (SD), RMSE, and $\rho$ as evaluation metrics. In general, lower values of MAE, RMSE, and SD indicate better performance (i.e., lower estimation error), while higher values of $\rho$ (closer to 1) reflect stronger correlation between predictions and ground truth.

Let $\mathbf{Y}_{pred}$ denote the predicted signal, $\mathbf{Y}_{gt}$ denote the ground truth signal, and $N$ be the total number of evaluation instances. The definitions of the adopted metrics are as follows:

**Mean Absolute Error (MAE):** It measures the average magnitude of the absolute differences between predicted and ground truth values, reflecting the overall prediction error without considering its direction.

$$\mathbf{MAE} = \frac{1}{N} \sum_{n=1}^{N} \left| \mathbf{Y}_{gt}^n - \mathbf{Y}_{pred}^n \right|. \tag{59}$$

**Root Mean Square Error (RMSE):** It evaluates the square root of the mean squared error, emphasizing larger errors due to the squaring operation, and is more sensitive to outliers than MAE.

$$\mathbf{RMSE} = \sqrt{\frac{1}{N} \sum_{n=1}^{N} \left( \mathbf{Y}_{gt}^n - \mathbf{Y}_{pred}^n \right)^2}. \tag{60}$$

**Standard Deviation (SD):** It quantifies the dispersion of the prediction errors around their mean, providing insight into the consistency and reliability of the predictions.

$$\mathbf{SD} = \sqrt{\frac{1}{N} \sum_{n=1}^{N} \left( \mathbf{Y}_e^n - \overline{Y}_e \right)^2}, \tag{61}$$

where the error term is defined as $\mathbf{Y}_e^n = \mathbf{Y}_{pred}^n - \mathbf{Y}_{gt}^n$, and $\overline{Y}_e$ denotes the mean error across all $N$ samples.

**Pearson's Correlation Coefficient ($\rho$):** It assesses the linear relationship between predicted and ground truth values, with higher values indicating stronger correlation and better temporal alignment.

$$\boldsymbol{\rho} = \frac{\sum_{n=1}^{N} (\mathbf{Y}_{gt}^n - \overline{\mathbf{Y}}_{gt})(\mathbf{Y}_{pred}^n - \overline{\mathbf{Y}}_{pred})}{\sqrt{\sum_{n=1}^{N} (\mathbf{Y}_{gt}^n - \overline{\mathbf{Y}}_{gt})^2 \sum_{n=1}^{N} (\mathbf{Y}_{pred}^n - \overline{\mathbf{Y}}_{pred})^2}}, \tag{62}$$

where $\overline{\mathbf{Y}}_{gt}$ and $\overline{\mathbf{Y}}_{pred}$ are the sample means of the ground truth and predicted signals, respectively.

### F.3    IMPLEMENTATION DETAILS.

The proposed FrePhys is implemented in PyTorch using the Adam optimizer. We train our model for 50 epochs, and the initial learning rate decay is 1e-3 with a shrink factor of 0.5 after every 5 epochs. The layer of the denoising network $L$ is 4, and the feature dimension $D$ is set to 64. For hyperparameters of the diffusion model, we follow PhysDiff's (Qian et al., 2025) setting. The maximum diffusion timestep $K$ is set to 1000. All experiments are performed on four NVIDIA GeForce RTX 4090 24G GPUs. The source code is available online.

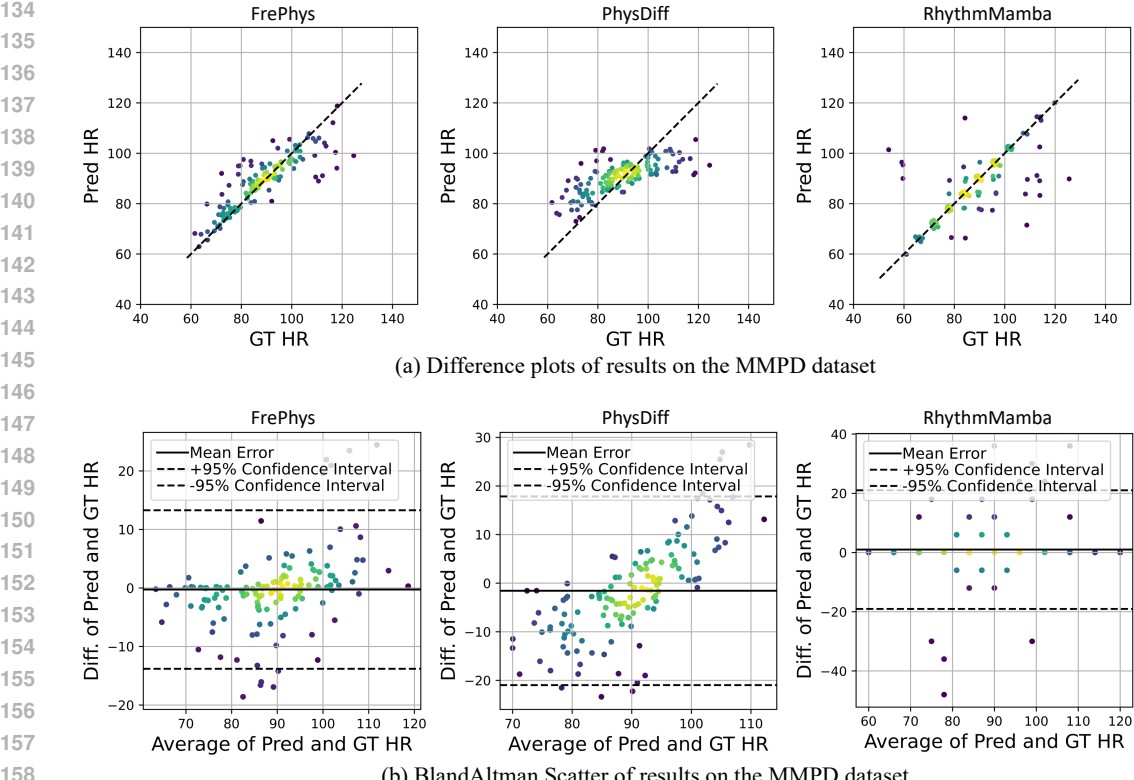

(a) Difference plots of results on the MMPD dataset

(b) BlandAltman Scatter of results on the MMPD dataset

Figure 7: Difference and BlandAltman scatter plots of results on the MMPD dataset

Table 6: Comparison of different bandpass filters on VIPL-HR. The computational cost test is conducted using a 10-second inference test for all filters on a single NVIDIA 4090 24GB GPU, reporting the throughput, inference time, and parameters. PBF achieves the best runtime efficiency while maintaining comparable performance. Best results are marked in bold.

| Filter | MAE↓ | RMSE↓ | $\rho$↑ | Parameters (M)↓ | Flops (G)↓ | Throughput (Kps)↑ | Inference time (ms)↓ | Memory (M)↓ |
|---|---|---|---|---|---|---|---|---|
| Butterworth | 3.75 | **6.28** | **0.86** | **0.86** | **7.75** | 83.84 | 11.93 | **874** |
| Chebyshev | **3.79** | 6.32 | **0.86** | **0.86** | **7.75** | 83.79 | 11.93 | **874** |
| Bessel | 3.52 | 6.46 | 0.85 | **0.86** | **7.75** | 80.59 | 12.41 | **874** |
| **PBF (ours)** | 3.79 | 6.34 | **0.86** | **0.86** | **7.75** | **85.44** | **11.70** | **874** |

# G    ADDITIONAL EXPERIMENTAL RESULTS

## G.1    IMPACT OF PHYSIOLOGICAL BANDWIDTH

As shown in Table 5, we investigated the impact of different physiological bandwidths on performance. The results indicate that the [0.66, 3.0] Hz range yields the best performance. In contrast, narrower bandwidths lead to performance degradation due to the loss of relevant physiological frequency information.

## G.2    IMPACT OF BANDWIDTH FILTER

In addition to our Physiological Bandpass Filter (PBF), we also explored soft filter alternatives. Specifically, we replaced the hard bandpass setup of PBF with flexible soft filters, including Butterworth, Chebyshev, and Bessel designs. These soft filters were implemented as non-parametric binary masks in the frequency domain, enabling smoother frequency responses.

As summarized in Tab. 6, all soft filters share the same parameter size. Among them, the Butterworth filter achieved slightly better accuracy than Chebyshev and Bessel, while the overall performance gains remained marginal. Importantly, our PBF consistently demonstrated the fastest infer-

Table 7: Ablation study on the number of cross-attention layers in the Cross-domain Representation Learning module, showing the trade-off between accuracy and inference cost on VIPL-HR. Best results are marked in bold.

| Layers | MAE↓ | RMSE↓ | $\rho$↑ | Parameters (M)↓ | Flops (G)↓ | Throughput (Kps)↑ | Inference time (ms)↓ | Memory (M)↓ |
|---|---|---|---|---|---|---|---|---|
| 1 | 7.42 | 10.60 | 0.66 | **0.27** | **1.97** | **401.68** | **2.49** | **872** |
| 2 | 4.84 | 7.82 | 0.81 | 0.47 | 3.40 | 200.36 | 4.99 | **872** |
| 3 | 3.90 | 6.52 | **0.86** | 0.66 | 5.82 | 127.58 | 7.84 | 874 |
| **4** | **3.79** | **6.34** | **0.86** | 0.86 | 7.75 | 85.44 | 11.70 | 874 |
| 5 | 3.97 | 6.60 | 0.85 | 1.06 | 9.68 | 75.74 | 13.20 | 876 |
| 6 | 5.19 | 7.74 | 0.82 | 1.26 | 11.61 | 62.24 | 16.07 | 876 |

ence speed and highest throughput. This suggests that while soft filters provide a valid alternative, the simplicity and efficiency of PBF make it a more practical choice for real-time rPPG applications.

### G.3 IMPACT OF CROSS-ATTENTION LAYERS

For the layers of cross-attention, we conducted ablations to analyze the optimal empirical setting of the cross-attention layer number. As shown in Table 7, performance improves with the layer number up to a certain point (4 layers), after which diminishing returns are observed along with increased inference cost. We thus adopt 4 layers as a balanced parameter setting.

### G.4 COMPUTATIONAL COST

About testing deployment feasibility, we conducted 10-second inference tests on a single NVIDIA RTX 4090 GPU (24GB), and report the key computational metrics below. As shown in the Table 8, our method offers significantly lower computational overhead than existing SOTA methods.

Table 8: Computational cost. The computational cost test is conducted using a 10-second inference test for all methods on a single NVIDIA 4090 24GB GPU. Best results are marked in bold and the second best in underline.

| Method | Parameters (M)↓ | Flops (G)↓ | Throughput (Kps)↑ | Inference time (ms)↓ | Memory (M)↓ |
|---|---|---|---|---|---|
| DeepPhys (Chen & McDuff, 2018) | 1.98 | 111.67 | 28.89 | 34.61 | 10638 |
| PhysNet (Yu et al., 2019) | **0.77** | 65.74 | 68.73 | 14.55 | 3750 |
| TS-CAN (Liu et al., 2020) | 1.98 | 111.67 | 26.23 | 38.13 | 11834 |
| PhysFormer (Yu et al., 2022) | 7.38 | 47.44 | 50.79 | 19.69 | 6480 |
| EfficientPhys (Liu et al., 2023) | 1.91 | 56.06 | 41.36 | 24.18 | 7814 |
| RhythmMamba (Zou et al., 2025b) | 2.00 | 12.41 | 27.16 | 36.82 | 2450 |
| PhysDiff (Qian et al., 2025) | 2.64 | 22.46 | 60.23 | 16.60 | 1246 |
| **FrePhys (ours)** | 0.86 | **7.75** | **85.44** | **11.70** | **874** |

### G.5 ADDITIONAL QUALITATIVE RESULTS FOR CONSISTENCY.

To further evaluate the consistency between predicted HR and ground truth measurements across different ranges, Figure 7 presents both scatter and BlandAltman plots on the MMPD dataset.

## H LIMITATIONS AND FUTURE WORK

Although our FrePhys shows strong robustness against general motion, it may still be susceptible to unseen highly dynamic or non-repetitive motion artifacts, especially those overlapping with the physiological frequency band. Future work could explore motion disentanglement techniques or multimodal fusion (e.g., depth or NIR sensors) to further enhance motion robustness. Besides, Diffusion models typically require multiple denoising iterations during inference, making them computationally heavier than traditional regression-based models. While we adopt acceleration techniques, real-time deployment on resource-constrained devices remains challenging.

# I  LLM USAGE

Large Language Models (LLMs) were used solely to assist with writing and polishing this manuscript. In particular, we employed an LLM to refine language, improve readability, and enhance the overall clarity and flow of the text (e.g., grammar checking, sentence rephrasing). The LLM was not involved in ideation, research methodology, experimental design, or data analysis. All scientific concepts, contributions, and results presented in this paper were conceived and executed entirely by the authors. The role of the LLM was strictly limited to improving linguistic quality.

The authors take full responsibility for the entire content of the manuscript, including any sections refined with LLM assistance. All usage adhered to ethical standards, and no plagiarism or scientific misconduct was introduced.

