# OpenReview forum: "FrePhys: Frequency-aware Diffusion Model for Remote Physiological Measurement"
_ICLR.cc/2026/Conference — ICLR 2026 Conference Withdrawn Submission_

### Official Review · Reviewer_x3v9 · 2025-10-26

**Soundness:** 3
**Presentation:** 2
**Contribution:** 3
**Rating:** 4
**Confidence:** 4

**Summary:**

This paper proposes to enhance the diffusion based rPPG deblurring by applying spectral priori to cross attention, for which 3 filters are desgined, namely, PBF, PSM, and ASS. PBF removes the spectral components outside the known band centering on the HR. PSM applies a hyper network to produce the mask at each frequency point, and ASS highlights the peak components in the spectrum by applying a given threshold. The experiments on 4 benchmarks show SOTA performance. The paper also includes mathematical proof on the uncertainy reduction caused by introduing the spectral proiri as guidance for denoising network in terms of entropy.

**Strengths:**

Spectral priori is introduced in refining the diffusion process. The experiment results seem good.

**Weaknesses:**

Why can the proposed innovations work is still not straightforward. The presentation could be improved.

**Questions:**

The implementation of the denoising network is not mentioned. It seems Transformer due to the cross attention in Fig. 2. Yet, I cannot understand why the filtered spectrum can help refining the cross attention. You concate the MSTmap and BVP as input to diffusion, and filter MSTmap to produce the outside control to the cross attention. Can you explain why?

Why use the same QK resulting from the so-called Physiological Frequency Denoiser for two different cross attentions, say, space-frequency and time-frequency cross-attention in Fig. 2? How do you define them? The two items seem heterogeneous, and cannot be tuned using the same QK in general.

In Fig. 2. why PBF is performed more than once? Band-pass filtering can be ensured by performing only once.

The aforementioned doubts can be clarified through a couple of abaltion studies to alter input to diffusion as either BVP or MSTmap, different QK for time and frequency respectively, and applying PBF only once.

What is the sensitivity of the hyper parameter Tao in the ASS? Any testing result?

The paper claims that the proposed method can capture both the time-domain transient and the global profile in frequency-domain. Such kind of claim should be cautious unless there is experimental evidence.

Theorem 1 is well known in the textbook of signal procesing. Proof in the appendix is unnecessary.

Symbol C appears all over the paper but with different meanings. Please add definitions or clarify via statement.

---

### Official Review · Reviewer_QuCX · 2025-10-31

**Soundness:** 2
**Presentation:** 3
**Contribution:** 2
**Rating:** 2
**Confidence:** 5

**Summary:**

This paper introduces FrePhys, a frequency-aware diffusion model for remote photoplethysmography (rPPG) — the task of estimating cardiac-related physiological signals from face videos. Existing diffusion-based rPPG methods (e.g., PhysDiff, DiffPhys) operate purely in the time domain, which struggles with irregular noise due to motion or illumination. FrePhys instead explicitly incorporates physiological frequency priors into the denoising process, thereby unifying spectral and temporal reasoning for robust pulse estimation.

**Strengths:**

- Frequency-domain conditioning: FrePhys is the first diffusion-based rPPG model to explicitly embed physiological frequency priors (0.66–3 Hz band, dominant harmonic enhancement) directly into the diffusion process.

- Clear physiological motivation: Grounded in cardiac spectral regularities: dominant peaks and narrowband energy, yielding interpretable and physiologically meaningful denoising.

- Empirical performance: Consistently surpasses state-of-the-art baselines on intra- and cross-dataset tests for HR, HRV, and respiration rate, showing excellent robustness to motion and illumination.

**Weaknesses:**

- Conceptual novelty moderate: The proposed modules (bandpass filter, modulation, thresholding) are adaptations of classical DSP principles; the main innovation lies in integration with diffusion.

- Computational complexity: Frequency transforms and cross-domain attention add overhead; runtime and latency are not benchmarked, which is relevant for real-time rPPG.

- Limited generalization study: Evaluation focuses on facial datasets; no validation on diverse demographics or extreme lighting conditions beyond VIPL-HR subsets. However, the CVPR 2022 “Synthetic Generation of Face Videos with Plethysmograph Physiology” dataset (Wang et al.) already provides a controlled yet demographically diverse benchmark that could be leveraged for fairer cross-subject evaluation.

- FrePhys uses a diffusion model as a learned iterative denoiser for rPPG, leveraging its probabilistic noise modeling and progressive refinement ability.
However, much of its success appears to come from the frequency-aware conditioning rather than diffusion itself. The diffusion backbone provides a principled yet computationally heavy scaffold; a simpler deterministic model with spectral priors might achieve similar results if trained carefully.

**Questions:**

- Minor typos and formatting issues (“learnig,” “ans future work”) and overlong mathematical appendices.
- Ethical statement could discuss bias mitigation for different skin tones or camera sensors.

---

### Official Review · Reviewer_hn9t · 2025-11-01

**Soundness:** 3
**Presentation:** 4
**Contribution:** 3
**Rating:** 8
**Confidence:** 2

**Summary:**

his paper proposes FrePhys, a frequency-aware diffusion model for remote photoplethysmography (rPPG) that addresses the vulnerability of existing time-domain methods to motion artifacts and illumination changes by integrating physiological frequency priors. The model first employs a three-stage physiological frequency denoiser (PBF for out-of-band noise suppression, PSM for enhancing cardiac harmonics, and ASS for adaptive in-band noise removal), then fuses frequency-domain insights with time-domain features via a cross-domain representation learning module, and finally reconstructs high-fidelity rPPG signals through a frequency-aware conditional diffusion process. Extensive experiments on four public datasets (UBFC-rPPG, PURE, MMPD, VIPL-HR) demonstrate that FrePhys outperforms state-of-the-art methods in heart rate (HR), heart rate variability (HRV), and respiration frequency (RF) estimation, especially under challenging motion conditions and cross-domain scenarios, while maintaining low computational cost.

**Strengths:**

The paper introduces a novel and effective paradigm by seamlessly embedding physiological frequency priors into diffusion modeling, which fills the gap of underutilizing frequency-domain insights in existing rPPG diffusion methods. The hierarchical denoising design (PBF, PSM, ASS) strategically targets different types of noise, enabling robust separation of physiological signals from interference in both out-of-band and in-band frequency ranges. The comprehensive experimental evaluation, covering multiple datasets, diverse physiological indicators, and cross-domain generalization, provides strong evidence for the model’s accuracy, robustness, and practical applicability.

**Weaknesses:**

1. FrePhys may still be susceptible to unseen highly dynamic or non-repetitive motion artifacts, especially those overlapping with the physiological frequency band [0.66, 3.0] Hz, which is explicitly acknowledged as a limitation in the paper and indicates room for improvement in handling such complex motion interference.
2. Despite the adoption of accelerated training and sampling techniques, diffusion models inherently require multiple denoising iterations during inference, making FrePhys computationally heavier than traditional regression-based models and posing challenges for real-time deployment on resource-constrained devices.
3. The paper’s experimental validation is mainly conducted on four standard public datasets (UBFC-rPPG, PURE, MMPD, VIPL-HR), but it fails to fully explore the model’s adaptability to diverse special populations (e.g., children or the elderly with different physiological frequency ranges) or extreme real-world lighting conditions, leaving the model’s versatility in broader practical scenarios insufficiently verified.

**Questions:**

See weakness

---

### Official Review · Reviewer_h8F5 · 2025-11-01

**Soundness:** 2
**Presentation:** 2
**Contribution:** 2
**Rating:** 2
**Confidence:** 4

**Summary:**

The paper propose a frequency-aware diffusion model for remote PPG estimation from face videos. Results on UBFC, PURE, VIPL-HR, and MMPD show state-of-the-art heart rate and HRV performance.

**Strengths:**

- We can see strong empirical performance and broad evaluation across four public datasets.
- The architecture is good, there is a clear problem framing, and thorough experiments.
- The frequency-domain motivation is sound and supported by good visual analysis.
- Cross-dataset tests and HRV/RF metrics are nice to have.
- There is sufficient figures, ablations, reproducibility, dataset coverage which makes paper easy to read and contributions well articulated.

**Weaknesses:**

- The novelty of the paper is limited. Frequency filtering, spectral masking, and Fourier-domain losses are well-explored in rPPG and signal processing. Embedding them inside a diffusion framework is logical, but not very theoretical or conceptual per se.
- The work is heavily on the engineering side and module stacking rather than a new principle that is theoretically grounded.
- Some claims (“rethinks how frequency information is used in rPPG”) are probably too much of a hype. In reality, what we have is a straightforward pipeline extension.
- The paper does not deeply analyze why diffusion + frequency priors beat simpler architectures. The results hence are mostly empirical.
- There is a little heavy compute and training complexity. On the other side, even with heavy complexity, there is a less conceptual advancement.
- It is not clear whether diffusion is needed or not here. A non-diffusion frequency-aware baseline could also close the gap?

**Questions:**

- Could a simpler time-freq network (e.g., transformer + spectral attention + freq-loss) achieve similar gains?
- Is the bandpass + adaptive mask really learnable necessity, or is this hand-designed DSP with training bolted on?
- Why diffusion versus denoising autoencoder or score-based model? What unique benefit does diffusion provide?
- How sensitive is the approach to sampling rate, ROI extraction steps, or dataset frequency bias?

---

### Note · Authors · 2025-11-13

I have read and agree with the venue's withdrawal policy on behalf of myself and my co-authors.